# Image Reconstruction Via Autoencoding Sequential Deep Image Prior

**Ismail R. Alkhouri**[*1,2], **Shijun Liang**[*3],
**Evan Bell**[1], **Qing Qu**[2], **Rongrong Wang**[1,4], **Saiprasad Ravishankar**[1,3]

[1]Department of Computational Mathematics, Science, & Engineering, Michigan State University
[2]Department of Electrical Engineering & Computer Science, University of Michigan - Ann Arbor
[3]Department of Biomedical Engineering, Michigan State University
[4]Department of Mathematics, Michigan State University
{alkhour3,liangs16,belleva1,wangron6,ravisha3}@msu.edu
{ismailal,qingqu}@umich.edu

## Abstract

Recently, Deep Image Prior (DIP) has emerged as an effective unsupervised one-shot learner, delivering competitive results across various image recovery problems. This method only requires the noisy measurements and a forward operator, relying solely on deep networks initialized with random noise to learn and restore the structure of the data. However, DIP is notorious for its vulnerability to overfitting due to the overparameterization of the network. Building upon insights into the impact of the DIP input and drawing inspiration from the gradual denoising process in cutting-edge diffusion models, we introduce Autoencoding Sequential DIP (aSeqDIP) for image reconstruction. This method progressively denoises and reconstructs the image through a sequential optimization of network weights. This is achieved using an input-adaptive DIP objective, combined with an autoencoding regularization term. Compared to diffusion models, our method does not require training data and outperforms other DIP-based methods in mitigating noise overfitting while maintaining a similar number of parameter updates as Vanilla DIP. Through extensive experiments, we validate the effectiveness of our method in various image reconstruction tasks, such as MRI and CT reconstruction, as well as in image restoration tasks like image denoising, inpainting, and non-linear deblurring. Our code is available at the GitHub repository aSeqDIP.

## 1 Introduction

Inverse imaging problems arise across various real-world applications [1]. These include tasks such as image denoising [2], image deblurring [3], restoring missing portions of images [4] (in-painting), super-resolution [5], magnetic resonance imaging (MRI) [6], and reconstructing X-ray computed tomography (CT) scans [7]. The common challenge in these scenarios is the reconstruction of images from limited and/or corrupted measurements.

Recently, a plethora of Deep Neural Network (DNN) techniques have emerged to solve inverse imaging problems, including supervised models that are trained to process measurements and generate estimates approximating the true image [8, 9, 10, 11, 12]. Generative model, such as Diffusion models (DMs) [13], have shown state-of-the-art (SOTA) performance in various imaging inverse problems such as the works in [14, 15, 16]. However, training diffusion models is not only computationally expensive but also demands a significant amount of clean (or fully-sampled) data points [17], limiting

---

[*]Equal contribution.

38th Conference on Neural Information Processing Systems (NeurIPS 2024).

their application to data-limited or data-corrupted tasks such as in medical imaging (MRI and CT). To address this challenge, previous DM-based MRI and CT methods typically fine-tune pre-trained DMs (trained on natural images) using limited fully-sampled data points, which are not as readily available as natural images [14, 15]. As such, there is a pressing need to develop methods that can alleviate the reliance on large, clean datasets and/or pre-trained models.

One well-known method is Deep Image Prior (DIP) [18], which does not require pre-trained models and relies on the parameters of an architecture (e.g., U-Net [19]). DIP finds a solution without needing any data other than the limited/corrupted task-specific measurements and the forward operator. DIP has shown competitive reconstruction/restoration results in inverse imaging problems such as MRI [20] and image denoising [18]. However, DIP (and most of its variants) suffer from the problem of overfitting.

In this paper, we introduce Autoencoding Sequential DIP (aSeqDIP) to mitigate noise overfitting artifacts. Our approach comprises a sequential update of network weights. A significant advantage of our approach, in contrast to DM-based methods, is its ability to operate without the need for training data or pre-trained models, thus fitting into the framework of *dataless* training [21]. Notably, the total number of parameter updates in aSeqDIP remains equivalent to that of Vanilla DIP [18]. Furthermore, unlike a recent work [20], aSeqDIP eliminates the need for gradient-based input updates.

**Contributions:**

- Building upon an insight about the impact of the DIP network input, we introduce Autoencoding Sequential DIP (aSeqDIP), which incorporates a U-Net architecture whose weights are updated sequentially. These updates are based on objective functions that consist of (*i*) an input-adaptive data consistency term and (*ii*) an autoencoding regularization term used for noise overfitting mitigation.
- Our extensive experimental evaluations, in terms of standard image reconstruction metrics and required run-time, highlight the superior (or competitive) performance of aSeqDIP compared to DIP-based and leading DM-based methods for the tasks of MRI and CT reconstruction, denoising, in-painting, and non-linear deblurring.

**Organization:**   In Section 2, we provide an overview of the considered tasks, and review recent advancements in both DIP-based and DM-based methods. Section 3 introduces our approach, aSeqDIP. Our experimental results are presented in Section 4, followed by discussions on conclusions and future directions in Section 5.

## 2   Preliminaries & Related Work

### 2.1   The Image Reconstruction Problem

Image reconstruction tasks can be defined as recovering an image $\mathbf{x}^* \in \mathbb{R}^n$ from measurements $\mathbf{y} \in \mathbb{R}^m$, where $m \leq n$, governed by the forward operator $\mathbf{A}$. For multi-coil MRI, the forward operator is $\mathbf{A} = \mathbf{MFS}$, where $\mathbf{M}$ denotes coil-wise undersampling, $\mathbf{F}$ is the coil-by-coil Fourier transform, and $\mathbf{S}$ represents the sensitivity encoding with multiple coils[2]. For CT, a simplified forward operator will be used to study the sparse-views setting: $\mathbf{A} = \mathbf{CR}$, where $\mathbf{C}$ is an undersampling operator that selects specific projection views or angles, and $\mathbf{R}$ denotes the radon transform [22] (corresponding to parallel-beam geometry in CT). For both MRI and CT with limited acquired data (done to accelerate scan or reduce X-ray dose, etc.), the condition $m < n$ applies.

For the denoising task, the forward operator is the identity matrix, whereas for in-painting, it is equivalent to applying a binary mask element-wise that masks corrupted or missing pixels. For deblurring, the non-linear forward operator is the neural network approximated kernel in [3]. Next, we review prior arts within the context of DIP-based and DM-based methods.

### 2.2   Related Work

**DIP-based Methods:**   Deep Image Prior (DIP) was first introduced by [18]. The authors demonstrated that the architecture of a generator network alone is capable of capturing a significant amount

---

[2]In practice, the entries of $\mathbf{x}^*$ and $\mathbf{y}$ in MRI are complex numbers. However, to generalize the problem's definition, we use real numbers.

of low-level image statistics even before any learning takes place. Specifically, the DIP image reconstruction is obtained through the minimization of the following objective:

$$\hat{\theta} = \underset{\theta}{\mathrm{argmin}} \ \|\mathbf{A}f_\theta(\mathbf{z}) - \mathbf{y}\|_2^2 \,, \quad \hat{\mathbf{x}} = f_{\hat{\theta}}(\mathbf{z}) \,, \tag{1}$$

where $\hat{\mathbf{x}}$ is the reconstructed image, and $\theta$ corresponds to the parameters of network $f : \mathbb{R}^n \to \mathbb{R}^n$, which is typically implemented using a U-Net architecture [19]. The input to the network, $\mathbf{z} \in \mathbb{R}^n$, is randomly chosen and remains fixed throughout the optimization process. While standard DIP was shown to perform well in many tasks, selecting the number of iterations to optimize objective (1) poses a challenge as the network would eventually fit the noise present in $\mathbf{y}$ or could fit to undesired images based on the null space of $\mathbf{A}$.

To mitigate the problem of noise overfitting, previous studies considered different approaches such as regularization, early stopping (ES), and network pruning [23]. For regularization-based methods, the work in [24] enhanced the standard DIP by introducing a total variation (TV) regularization term for denoising and deblurring tasks, whereas the study in [25] proposed combining DIP with stochastic gradient Langevin dynamics (SGLD) [26]. The authors in [27] use running variance as the criterion for ES, whereas the authors of [28] propose combining self-validation and training to apply ES.

The input to the standard DIP (or Vanilla DIP) network is a random noise vector that, in most works, remains fixed during the optimization. Nevertheless, other works, such as those in [29] and [30], have explored cases where the input contains some structure of the ground truth. The approach employed in reference-guided DIP (Ref-Guided DIP) [29] follows the same objective as standard DIP in (1). However, instead of using a fixed random noise vector as input, it utilizes a reference image closely resembling the one undergoing reconstruction. This method was applied to the task of MRI. This methodology proves particularly effective when datasets comprising structurally similar data points are available. The reference required here makes this method a data-dependent approach.

Inspired by the departure from using a random fixed input, the authors in [20] recently introduced Self-Guided DIP. Unlike Ref-Guided DIP [29], a prior image that closely resembles the unknown (to be estimated) image is not needed, and the optimization occurs simultaneously with respect to both the input and the parameters of the network. Specifically, Self-Guided DIP employs the following objective:

$$\hat{\theta}, \hat{\mathbf{z}} = \underset{\theta, \mathbf{z}}{\mathrm{argmin}} \|\mathbf{A}\mathbb{E}_{\boldsymbol{\eta}}[f_\theta(\mathbf{z} + \boldsymbol{\eta})] - \mathbf{y}\|_2^2 + \alpha\|\mathbb{E}_{\boldsymbol{\eta}}[f_\theta(\mathbf{z} + \boldsymbol{\eta})] - \mathbf{z}\|_2^2 \,, \tag{2}$$

where $\boldsymbol{\eta}$ is random noise, and $\alpha$ is a regularization parameter. The first (resp. second) term is used for data consistency (resp. denoising regularization) and final reconstruction is obtained as $\hat{\mathbf{x}} = \mathbb{E}_{\boldsymbol{\eta}}[f_{\hat{\theta}}(\hat{\mathbf{z}} + \boldsymbol{\eta})]$. aSeqDIP is different from Self-Guided DIP as our method does not require gradient-based updates for the input, making it computationally less expensive. Self-Guided DIP has demonstrated superior performance compared to Vanilla DIP, TV-DIP, and SGLD-DIP, thus serving as a primary baseline for comparison.

**DM-based Methods:** In recent years, there has been an abundance of DM-based methods proposed to address inverse imaging problems [14, 15, 16, 31, 32, 33]. A well-known method for natural images is Diffusion Posterior sampling (DPS) [16]. DPS incorporates a gradient step into the reverse sampling process of pre-trained DMs, ensuring data consistency and enabling sampling from the conditional distribution. In the context of image reconstruction and restoration tasks, numerous diffusion-based approaches have emerged, as evidenced by works such as [34, 35, 36]. Notably, the authors in [14] and [15] introduced a SOTA DM-based approach for addressing the MRI and CT reconstruction inverse problems, respectively. They propose incorporating the predictor-corrector sampling algorithm [37] for data consistency, akin to DPS, thereby facilitating the sampling from a conditional distribution.

One clear distinction between aSeqDIP and DM-based methods is that our approach does not necessitate pre-trained models. For our experiments in MRI, CT, and denoising (as well as in-painting and deblurring) tasks, we will utilize Score-MRI [14], Manifold Constrained Gradient (MCG) [15], and DPS [16], respectively, as DM-based baselines.

# 3 Autoencoding Sequential DIP

In this section, we begin by investigating the impact of the input on DIP. Then, we introduce our method, aSeqDIP. We note that while we consider linear and non-linear inverse problems, in our formulations, we use a linear forward model to simplify notation.

## 3.1 Motivation of aSeqDIP: The Impact of the Network Input in Vanilla DIP

Here, we aim to address the question: *How does employing a noisy version of the ground truth image, which retains some structure of the ground truth, as the fixed input to the Vanilla DIP objective in* (1), *affect performance?* To investigate, we conduct the following experiment.

Consider the MRI task defined as $\mathbf{y} \approx \mathbf{A}\mathbf{x}^*$. Let the input to the standard DIP objective in (1) be denoted as $\mathbf{z} = \mathbf{x}^* + \boldsymbol{\delta}$, where $\boldsymbol{\delta} \sim \mathcal{N}(\mathbf{0}, \sigma^2 \mathbf{I})$. Here, $\sigma$ controls the magnitude of the perturbations added to the ground truth image, indicating that a larger $\sigma$ results in a greater deviation between $\mathbf{z}$ and $\mathbf{x}^*$. We optimize (1) for various values of $\sigma$, recording the best possible PSNR compared to the ground truth, i.e., prior to the start of the noise overfitting decay.

Figure 1 displays the average results for 8 images. Notably, for all images, a closer similarity of the DIP network input to $\mathbf{x}^*$, as indicated by $\sigma$, corresponds to higher reconstruction quality, measured by PSNR. Larger variance in the standard Gaussian distribution corresponds to larger additive perturbations even for the case of $\mathbf{x}^* = \mathbf{0}$ (the red curve). We conjecture that this still leads to larger distances from the ground truth and hence worse performance.

Based on this discussion, a notable insight emerges:

> The proximity of the DIP network input to the ground truth correlates with the quality of the reconstruction. This promotes the question: *Can we develop an input-adaptive DIP method that mitigates noise overfitting?*

We proceed to address this question by proposing our method, which we refer to as Autoencoding Sequential DIP (aSeqDIP). In Appendix A, we provide a case study and theory on the impact of the DIP network input through the lens of the Neural Tangent Kernel in residual networks. The onset of severe noise overfitting therein is delayed for better inputs (Appendix A.2).

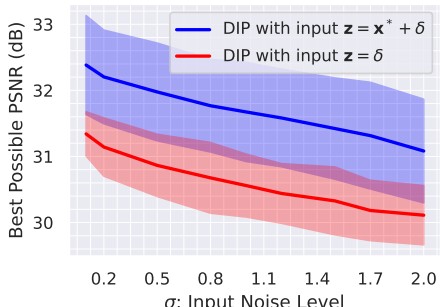

Figure 1: Average best possible PSNR values (in dB) obtained from standard DIP in (1) for 8 MRI (with 4x acceleration factor) scans (y-axis), where the network input $\mathbf{z}$ is either a perturbed version of the ground truth or pure noise. The noise is a zero-mean additive Gaussian noise with strength determined by $\sigma$ (x-axis).

## 3.2 The Proposed aSeqDIP Algorithm

Consider that we have a U-Net architecture defined by $f : \mathbb{R}^n \to \mathbb{R}^n$ whose weights are given by $\phi_k$, where $k \in [K]$, and $[K] := \{1, \ldots, K\}$. Each set of parameters in $f_{\phi_k}$ takes an input $\mathbf{z}_k$ and outputs $f_{\phi_k}(\mathbf{z}_k)$. Based on the insight from the previous subsection, we initially set $\mathbf{z}_0$ to $\mathbf{y}$ (resp. $\mathbf{A}^H \mathbf{y}$) for denoising, in-painting, and deblurring (resp. MRI and CT). The initialization of $\phi_1$ follows the same initialization as any other DIP-based method. The parameters in $f_{\phi_k}$, and the input, $\mathbf{z}_k$, are then updated sequentially through

$$\phi_k \leftarrow \underset{\phi_k}{\operatorname{argmin}} \|\mathbf{A}f_{\phi_k}(\mathbf{z}_{k-1}) - \mathbf{y}\|_2^2 + \lambda \|f_{\phi_k}(\mathbf{z}_{k-1}) - \mathbf{z}_{k-1}\|_2^2 , \tag{3}$$

$$\mathbf{z}_k \leftarrow f_{\phi_k}(\mathbf{z}_{k-1}) , \tag{4}$$

where $\lambda \in \mathbb{R}_+$ is a regularization parameter, and the initialization of $\phi_k$ is the optimized $\phi_{k-1}$ in (3). The final reconstruction is given as:

$$\hat{\mathbf{x}} = \mathbf{z}_K = f_{\phi_K}(\mathbf{z}_{K-1}) . \tag{5}$$

The proposed procedure outlined in (3) and (4) consists of two key components. First, the optimization of each set of weights in $f_{\phi_k}$ using an objective that consists of the data consistency term and the

---

**Algorithm 1** **A**utoencoding **Seq**uential **D**eep **I**mage **P**rior (**aSeqDIP**).

---

**Input**: Measurements $\mathbf{y}$, forward operator $\mathbf{A}$, number of input updates $K$, number of gradient updates $N$ per input update, regularization parameter $\lambda$, and learning rate $\beta$.

 **Output**: Reconstructed image $\hat{\mathbf{x}}$.

 **Initialization**: $\mathbf{z}_0 = \mathbf{A}^H \mathbf{y}$; $\phi_0 \sim \mathcal{N}(\mathbf{0}, \mathbf{I})$.

1: **For each** $k \in [K]$

2:    **Initialize** $\phi_k^{(0)} \leftarrow \phi_{k-1}^{(N)}$ for $k \in \{2, \ldots, K\}$, and $\phi_k^{(0)} \leftarrow \phi^{(0)}$ for $k = 1$.

3:    **For each** $i \in [N]$. (Network parameters update)

4:        $\phi_k^{(i)} = \phi_k^{(i-1)} - \beta \nabla_{\phi_k} \left[ \|\mathbf{A} f_{\phi_k}(\mathbf{z}_{k-1}) - \mathbf{y}\|_2^2 + \lambda \|f_{\phi_k}(\mathbf{z}_{k-1}) - \mathbf{z}_{k-1}\|_2^2 \right]\Big|_{\phi_k = \phi_k^{(i-1)}}$.

5:    **Obtain** $\mathbf{z}_k := f_{\phi_k^{(N)}}(\mathbf{z}_{k-1})$. (Network input update)

6: **Reconstructed image**: $\hat{\mathbf{x}} = \mathbf{z}_K = f_{\phi_K^{(N)}}(\mathbf{z}_{K-1})$

---

Figure 2: Illustrative block diagram of the proposed aSeqDIP procedure. Each trapezoid corresponds to the updates of $f_{\phi_k}$ that takes $\mathbf{z}_{k-1}$ as input and is initialized with the optimized parameters $\phi_{k-1}$ for $k \in \{2, \ldots, K\}$ or randomly for $k = 1$. The optimization for each set of weights takes place based on (3) and is run for $N$ steps. The final reconstruction is $f_{\phi_K}(\mathbf{z}_{K-1})$.

second autoencoding term that aims to alleviate noise overfitting. Second, the update of the input, $\mathbf{z}_k$, after optimizing each set of weights $f_{\phi_k}$, so that our method is *input-adaptive*.

Algorithm 1 presents the procedure of our proposed approach. As inputs, the algorithm takes $\mathbf{y}$, $\mathbf{A}$, $K$, $N$, $\lambda$, and the learning rate $\beta$. Apart from the measurements and the forward operator, the remaining parameters are considered hyper-parameters, typical in most DIP-based methods. The parameters in $f_{\phi_k}$ are set to $\phi_{k-1}$ (step 2) and subsequently optimized for $N$ iterations using a gradient-based optimizer, such as gradient descent (as depicted in Algorithm 1) or Adam [38]. A block diagram of our proposed aSeqDIP method is presented in Figure 2.

In the following remarks, we provide insights into our proposed aSeqDIP method.

**Remark 3.1** (Differences from Vanilla DIP [18])**.** Assume that the iterates of (3) and (4) converge, i.e., as $k \to \infty$, $\mathbf{z}_k \to \mathbf{z}^*$ and $\phi_k \to \phi^*$. Then, according to (4), for a continuous mapping $f$, we have $\mathbf{z}^* = f_{\phi^*}(\mathbf{z}^*)$. Substituting this into (3) in the limit, we get $\phi^* = \{\arg\min_\phi \|\mathbf{A} f_\phi(\mathbf{z}^*) - \mathbf{y}\|_2^2 : f_\phi(\mathbf{z}^*) = \mathbf{z}^*\}$, which corresponds to the minimizer of

$$\min_\phi \|\mathbf{A} f_\phi(\mathbf{z}) - \mathbf{y}\|_2^2 \quad s.t. \quad \mathbf{z} = f_\phi(\mathbf{z}). \tag{6}$$

The limit points of aSeqDIP correspond to the solution of a constrained version of the Vanilla DIP objective in (1). The constraint enforces additional prior that could alleviate overfitting. While its not straightforward to use a gradient-based algorithm for (6) given the hard constraint, the aSeqDIP scheme's limit points nevertheless minimize (6). Furthermore, aSeqDIP automatically estimates the network input by a sequential feed forward process without needing expensive updates. The main point is to show that aSeqDIP is solving the optimization problem in (6), which is different than Vanilla DIP in (1).

**Remark 3.2** (Differences from Self-Guided DIP [20])**.** While both aSeqDIP and Self-Guided DIP [20] update the input and network parameters simultaneously, there exist fundamental differences. Firstly, Self-Guided DIP solves the optimization problem in (2) which does not strictly enforce the auto-encoder constraint $\mathbf{z} = f_\phi(\mathbf{z})$ as in (6). Secondly, aSeqDIP only requires a network forward pass to update $\mathbf{z}$, resulting in significantly fewer computations as will further be demonstrated in our experimental results. Thirdly, the second term in the aSeqDIP objective does not require computing the expectation, as it is an auto-encoder rather than a denoiser that results in higher resistance to noise overfitting. Lastly, our method does not require initializing $\mathbf{z}$ randomly and generating random vectors ($\boldsymbol{\eta}$ in (2)). The selection of $\boldsymbol{\eta}$ introduces an additional hyper-parameter that we avoid, focusing solely

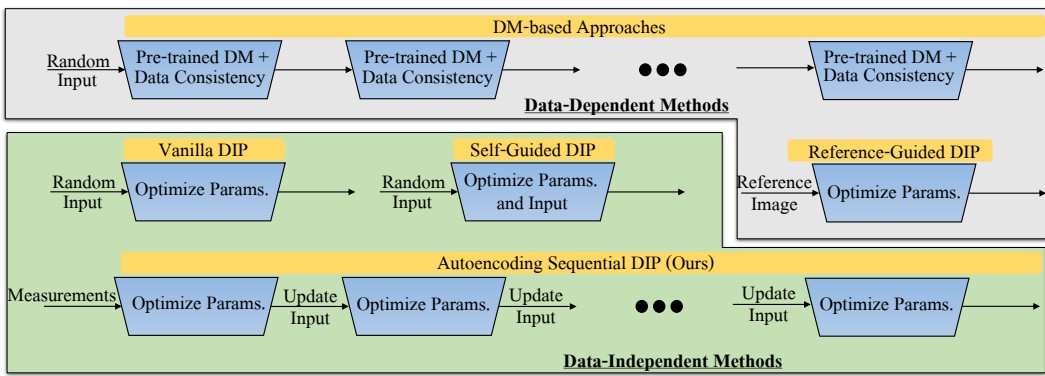

Figure 3: **An overview of differences between aSeqDIP and prior arts** in terms of data dependency, network architecture(s), and procedural requisites. '**Data-Dependency**' here indicates whether a method depend on a prior reference image or pre-trained models.

on selecting $NK$ (total number of iterations) and $\lambda$ (regularization strength), which are necessary in most DIP-based methods.

**Remark 3.3** (Computational Requirements). The computational requirements of aSeqDIP are determined by two factors: (*i*) the $NK$ gradient-based parameter updates, and (*ii*) the number of function evaluations necessary for updating $\mathbf{z}$, which is $K$. In our experiments, we have found that setting $N = 2$ and $K = 2000$ is generally sufficient. This configuration makes aSeqDIP nearly as efficient as Vanilla DIP.

**Remark 3.4** (Relationship to DMs). aSeqDIP bears resemblance to the reverse process in DMs due to their shared gradual denoising steps. However, despite these similarities, several distinctions emerge. Firstly, unlike the DM network, aSeqDIP does not require encoding a scalar representing time $t$. Secondly, and perhaps most significantly, aSeqDIP operates without requiring any training data or pre-trained networks. Thirdly, aSeqDIP operates in a truly sequential manner in terms of time, whereas in DMs, whether it's training (e.g., denoising score matching [39]) or sampling, the prevalent technique involves sampling from time $t \sim \mathcal{U}[0, 1]$ (uniform distribution), which allows for non-sequential time points.

Figure 3 illustrates how different approaches compare to aSeqDIP.

### 3.2.1 Mitigating Noise Overfitting in aSeqDIP

In DIP-based approaches, noise overfitting occurs as the network attempts to fit its output to the noisy or subsampled measurements, $\mathbf{y}$, as $k$ increases during training. However, the specific value of $k$ at which this PSNR decay begins is uncertain and varies across tasks and even among images within the same task and distribution. In aSeqDIP, when the output of network $f_{\phi_k}$ improves compared to that of $f_{\phi_{k-1}}$, the autoencoder term enforces similarity between the input and output of the network, thus delaying the onset of noise overfitting decay. This occurs because we are not only enforcing the network output to be measurement-consistent, but also enforcing that the output and input become similar. Consequently, as $k$ increases, noise fitting is delayed, and utilizing the autoencoder provides regularization against noise overfitting. In Section 4, we will demonstrate how the proposed autoencoding term effectively regulates noise overfitting.

One might expect that incorporating the autoencoder could negatively impact reconstruction quality. However, empirical observations reveal that not only is noise overfitting delayed with the autoencoder term, but also image reconstruction quality is enhanced. To further support this statement, in Appendix B, we investigate whether a trained autoencoder on clean images can act as a reconstructor at testing time by optimizing the input.

| Task | Setting | Data-independent baselines | Data-dependent baselines |
|------|---------|---------------------------|--------------------------|
| MRI | Ax $\in \{4x, 8x\}$ | Vanilla DIP [18], ES-DIP [27], TV-DIP [24], Self-Guided DIP [20] | Ref-Guided DIP [29] Score-MRI [14] |
| CT | views $\in \{18, 30\}$ | Vanilla DIP [18], Self-Guided DIP [20], Filter Back Projection (FBP) [42] | Ref-Guided DIP [29] MCG [15] |
| Denoising | $\sigma_{\mathrm{d}} \in \{15, 30\}$ | Vanilla DIP [18], ES-DIP [27], Self-Guided DIP [20], TV-DIP [24], Rethinking-DIP [43], SGLD-DIP [25] | DPS [16] |
| In-painting | HIAR $\in \{0.1, 0.25\}$ | Vanilla DIP [18], ES-DIP [27], Self-Guided DIP [20], SGLD-DIP [25], TV-DIP [24] | DPS [16] |
| Deblurring | BKSE [3] | Self-Guided DIP [20] SGLD-DIP [25] | DPS [16] |

Table 1: Tasks, settings, and baselines considered in our experiments. For MRI, we consider two Acceleration (Ax) factors, 4x and 8x, that determine the subsampling of the measurements. For 2D CT (parallel beam geometry), we use two sparse view settings: 18 and 30 views. For denoising, we perturb the ground truth images using two noise levels determined by $\sigma_{\mathrm{d}}$. In in-painting, we use two hole-to-image area ratios (HIAR), 0.1 and 0.25. For non-linear deblurring, we use the Blurring Kernel Space Exploring (BKSE) setting [3], described in Equations (56) to (59) of [16]. Each baseline that utilizes pre-trained models or a reference image is considered data-dependent. Further details are provided in Appendix C.6.

# 4 Experimental Results

## 4.1 Settings, Datasets, and Baselines

In our experiments, we consider five tasks: MRI reconstruction from undersampled measurements, sparse-view CT image reconstruction, denoising, non-linear deblurring and in-painting. For MRI, we use the fastMRI dataset[3]. The forward model is $\mathbf{y} \approx \mathbf{A}\mathbf{x}^*$. The multi-coil data is obtained using 15 coils and is cropped to a resolution of $320 \times 320$ pixels. To simulate undersampling of the MRI k-space, we use a Cartesian mask with 4x and 8x accelerations. Sensitivity maps for the coils are obtained using the BART toolbox [40]. For CT, we use the AAPM dataset[4]. For parallel beam CT, the input image with $512 \times 512$ pixels is transformed into its sinogram representation using a Radon transform (the operator $\mathbf{A}$). The forward model assuming a monoenergetic source and no scatter, noise is $y_i = I_0 e^{-[\mathbf{A}\mathbf{x}^*]_i}$, with $I_0$ denoting the number of incident photons per ray (assumed to be 1 for simplicity) and $i$ indexing the $i$th measurement or detector pixel. We use the post-log measurements for reconstruction. We use a full set of 180 projection angles and simulate two different sparse view acquisition scenarios (with equispaced angles). Specifically, we created cases with 18 and 30 angles/views. The image resolution is kept at a fixed size.

For the tasks of denoising, in-painting, and non-linear deblurring, we use the CBSD68 dataset[5]. For each task, we use 20 measurements/corrupted images. To evaluate the reconstruction quality, we use the Peak Signal to Noise Ratio (PSNR), and the Structural SIMilarity (SSIM) index [41]. For experimental settings and baselines, see Table 1 and its caption. Note that we consider data-dependent and data-independent baselines as shown in the third and fourth columns of Table 1. All the experiments are conducted on a single RTX5000 GPU machine. Further implementation details are provided in Appendix C.6.

For the proposed aSeqDIP method in Algorithm 1, we use the Adam optimizer with learning rate of $\beta = 0.0001$. Furthermore, the regularization parameter is set to $\lambda = 1$ following the ablation study in Appendix C.4. We select $N = 2$ and $K = 2000$ following the ablation study in Appendix C.5.

## 4.2 Impact of the Autoencoding term on Noise Overfitting

In this subsection, we showcase the impact of the proposed autoencoding regularization in aSeqDIP on noise or null space (nuisance) overfitting.

We conducted experiments using 20 MRI scans and 20 CT scans, considering two cases of aSeqDIP as outlined in Algorithm 1. The first case sets $\lambda = 1$, consistent with the remainder of the paper,

---

[3] https://github.com/microsoft/fastmri-plus/tree/main
[4] https://www.aapm.org/grandchallenge/lowdosect/
[5] https://github.com/clausmichele/CBSD68-dataset

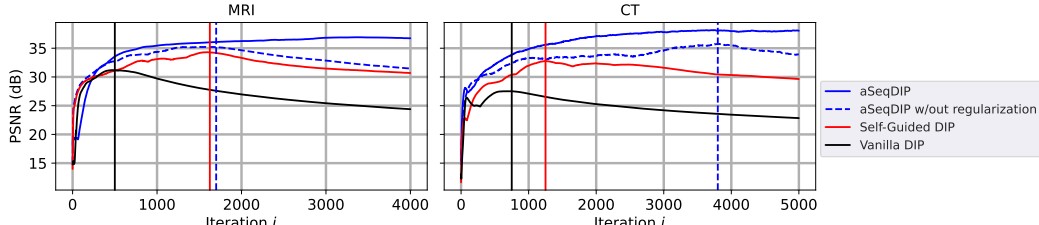

Figure 4: Average PSNR results w.r.t. iteration $i$ of 20 MRI (with 4x) scans (*left*) and 20 CT (with 18 views) scans (*right*) to show the impact of the proposed autoencoding regularization term on noise overfitting in aSeqDIP. Furthermore, average results of Vanilla DIP and Self-Guided DIP are also reported for comparison. For aSeqDIP, iteration $i \in [NK]$, where $N = 2$. Vertical lines approximately indicate the start of the PSNR decay for every case. In Appendix C.1, we include the PSNR curves of aSeqDIP and other DIP-based methods for the task of denoising.

while the second case sets $\lambda = 0$, effectively disabling the autoencoding regularization term in (3). Additionally, for comparison, we report results for Vanilla DIP and Self-Guided DIP. The average PSNR results for these cases are depicted in Figure 4.

As observed, when the autoencoder term is disabled in aSeqDIP (blue dashed lines), noise overfitting in MRI, akin to Self-Guided DIP, begins after nearly 1600 iterations. For CT, we note that aSeqDIP without regularization starts noise overfitting at around iteration 3800, whereas Self-Guided DIP experiences PSNR decay earlier, after approximately 1250 iterations. Importantly, when the autoencoding term is utilized (blue solid lines), not only does the decay in noise overfitting not commence until after iteration 4000, but the reconstruction quality (measured by PSNR) also improves.

As expected, PSNR decay in Vanilla DIP begins early, at around iteration 500 and 750 for MRI and CT, respectively. In Appendix C.4, we provide an ablation study to better show the impact of the value of $\lambda$ in aSeqDIP.

### 4.3 Main Results

Here, we present our primary results regarding the reconstruction quality, measured by PSNR and SSIM, as well as the associated run-time. Table 2 presents the results for the considered tasks in this paper. Column 3 indicates whether the baselines depend on prior data or pre-trained models. The last three columns provide the PSNR, SSIM, and run-time results where the arrows indicate favorable results. For PSNR and SSIM, the settings correspond to the second column of Table 1. The black (resp. blue) text corresponds to the first (resp. second) setting. Values after the $\pm$ sign indicate standard deviation. Subsequently, we offer observations on the main results.

Compared to data-independent methods, i.e., the baselines that do not depend on a reference image or pre-trained models, aSeqDIP demonstrates improved PSNR and SSIM scores. For example, aSeqDIP, apart from Self-Guided DIP, shows nearly a 1dB improvement for MRI 8X acceleration compared to conventional methods. For the task of 30-views CT, aSeqDIP reports SSIM score of 0.92 which is 5% more than the second best, which is Self-Guided DIP with SSIM of 0.872. Although improvements against Self-Guided DIP are generally marginal in terms of reconstruction quality, our method proves to be 2X faster for MRI and CT reconstruction and requires 1 minute less than Self-Guided DIP for denoising and in-painting. This speed-up is attributed to updating the input using one forward pass of the trained network at each iteration $k$, instead of computing gradients with respect to the input for the update. Compared to Vanilla DIP, our method, on average, only requires an additional 30 to 60 seconds. When compared to ES-DIP [27], our method requires longer time, but on average achieves better reconstruction results across three tasks and different settings.

In comparison to data-dependent methods such as Score-MRI and MCG, our approach not only yields the best PSNR and SSIM but also requires reduced run-time, all without requiring any training data or pre-trained models. For instance, on average, aSeqDIP achieves nearly a 2dB improvement in 30-views CT compared to MCG while being 2X faster. In comparison to DPS, on average, our method report higher SSIM. Our method requires slightly less run-time on average but enhances the PSNR by approximately 0.6dB for both denoising and in-painting. Notably, our method is an optimization-based approach, whereas DM-based methods only require function evaluations. However, the generally larger run-time reported for DM-based methods is due to the necessity of

| Task | Method | Data Independency | PSNR (dB) (↑) (Setting 1, Setting 2) | SSIM ∈ [0, 1] (↑) (Setting 1, Setting 2) | Run-time (↓) (minutes) |
|------|--------|-------------------|--------------------------------------|------------------------------------------|------------------------|
| MRI | Score-MRI | × | (31.51±0.45, 29.61±0.44) | (0.891±0.012, 0.862±0.014) | 6.2±0.12 |
| | Ref-Guided DIP | × | (33.17±0.27, 30.23±0.24) | (0.912±0.021, 0.873±0.016) | 2.5±0.2 |
| | TV-DIP | ✓ | (30.52±0.25, 29.20±0.37) | (0.872±0.022, 0.852±0.022) | 2.5±0.1 |
| | ES-DIP | ✓ | (31.02±0.34, 29.44±0.45) | (0.882±0.031, 0.858±0.028) | 1.56±0.34 |
| | Vanilla DIP | ✓ | (30.21±0.42, 28.75±0.33) | (0.865±0.02, 0.842±0.022) | **1.5±0.12** |
| | Self-Guided DIP | ✓ | (33.6±0.23, 30.75±0.25) | (0.922±0.008, 0.874±0.006) | 4.5±0.67 |
| | aSeqDIP (Ours) | ✓ | (**34.08±0.41**, **31.34±0.47**) | (**0.929±0.008**, **0.887±0.009**) | 2.2±0.12 |
| CT | MCG | × | (32.82±0.52, 31.35±0.49) | (0.912±0.08, 0.852±0.09) | 6.4±0.2 |
| | FBP | ✓ | (22.92±0.22, 19.52±0.32) | (0.75±0.021, 0.68±0.023) | **0.2±0.01** |
| | Ref-Guided DIP | × | (31.21±0.24, 28.31±0.42) | (0.892±0.023, 0.842±0.021) | 2.5±0.42 |
| | Vanilla DIP | ✓ | (26.21±0.12, 24.31±0.34) | (0.791±0.021, 0.772±0.012) | 1.5±0.21 |
| | Self-Guided DIP | ✓ | (33.95±0.32, 31.95±0.32) | (0.918±0.02, 0.872±0.031) | 4.5±0.56 |
| | aSeqDIP (Ours) | ✓ | (**34.88±0.36**, **33.09±0.39**) | (**0.941±0.026**, **0.92±0.022**) | 2.2±0.42 |
| Denoising | DPS | × | (31.02±0.25, 28.2±0.31) | (0.912±0.02, 0.882±0.021) | 2.5±0.17 |
| | Vanilla DIP | ✓ | (30.48±0.28, 27.84±0.32) | (0.905±0.021, 0.871±0.030) | **1.5±0.22** |
| | SGLD DIP | ✓ | (30.58±0.34, 28.12±0.42) | (0.908±0.021, 0.877±0.017) | 3.2±0.24 |
| | TV-DIP | ✓ | (30.57±0.31, 28.47±0.26) | (0.914±0.022, 0.882±0.014) | 2.5±0.24 |
| | Rethinking-DIP | ✓ | (30.98±0.31, 28.67±0.25) | (0.912±0.02, 0.887±0.03) | 2.5±0.34 |
| | ES-DIP | ✓ | (31.11±0.23, 28.12±0.41) | (0.914±0.017, 0.886±0.024) | 1.45±0.44 |
| | Self-Guided DIP | ✓ | (31.21±0.26, 28.31±0.35) | (0.916±0.02, 0.891±0.03) | 3.5±0.45 |
| | aSeqDIP (Ours) | ✓ | (**31.51±0.34**, **28.97±0.44**) | (**0.926±0.021**, **0.908±0.031**) | 2.4±0.45 |
| In-Painting | DPS | × | (23.9±0.45, 22.03±0.36) | (0.817±0.023, 0.762±0.021) | 2.5±0.3 |
| | Vanilla DIP | ✓ | (22.56±0.31, 21.32±0.67) | (0.754±0.023, 0.721±0.012) | **1.5±0.35** |
| | SGLD DIP | ✓ | (23.09±0.55, 21.41±0.45) | (0.772±0.023, 0.732±0.041) | 2.5±0.45 |
| | TV-DIP | ✓ | (22.87±0.45, 21.64±0.51) | (0.774±0.04, 0.742±0.042) | 2.5±0.31 |
| | ES-DIP | ✓ | (23.33±0.44, 21.89±0.28) | (0.781±0.034, 0.745±0.041) | 1.25±0.55 |
| | Self-Guided DIP | ✓ | (23.84±0.43, 21.78±0.52) | (0.792±0.042, 0.752±0.064) | 3.5±0.45 |
| | aSeqDIP (Ours) | ✓ | (**24.56±0.45**, **22.57±0.47**) | (**0.838±0.051**, **0.778±0.045**) | 2.4±0.54 |
| Deblurring | DPS | × | (23.40±0.56) | (0.776±0.032) | 2.24±0.65 |
| | SGLD DIP | ✓ | (19.80±0.43) | (0.720±0.03) | 3.24±0.55 |
| | Self-Guided DIP | ✓ | (20.34±0.55) | (0.732±0.025) | 3.4±1.02 |
| | aSeqDIP (Ours) | ✓ | (**23.89±0.40**) | (**0.792±0.033**) | 2.5±0.78 |

Table 2: Average PSNR, SSIM, and run-time results reported by our method against the selected baselines for the tasks of MRI reconstruction, CT reconstruction, image denoising, in-painting, and non-linear deblurring. '**Data-Independency**' in column 3 indicates whether the methods depend on prior data or pre-trained models. Setting 1 and Setting 2, in the fourth and fifth columns correspond to the scenarios in the second column of Table 1. For tasks with two settings, the run-time results are averaged over the two settings. Values past ± represent the standard deviation. See Appendix C.2 and Appendix C.3 for more comparison results.

running a large number of reverse sampling steps. When compared to Ref-Guided DIP, our method achieves higher PSNR and SSIM results without the need for any prior (or reference image) image.

## 4.4 Visualizations

Figure 5 shows reconstructed images for the five considered tasks using aSeqDIP and the other baselines. Each row corresponds to a task. The first column displays the ground truth (GT) image whereas the second column shows the degraded image. Column 3 to the column before last present the reconstructed images by the baselines, while the last column shows the reconstructed images by aSeqDIP. PSNR values are provided at the bottom of each reconstructed image.

As observed, aSeqDIP achieves the highest PSNR scores. Additionally, the top right green boxes, which show the difference between the central region of the reconstructed and GT images, indicate that for MRI and CT, our method visually exhibits the least difference, making it the closest to the GT.

A similar observation is seen for the denoising task for the zoomed in bottom box. For inpainting, we note that aSeqDIP introduces the fewest unwanted artifacts as observed in the clouds (for DPS), and the left wing of the plane. While aSeqDIP contains artifacts for the task of Deblurring when compared to DPS, the latter generates a perceptually different image when compared to the GT. Similar observation are noticed with the additional visualizations provided in Appendix C.7.

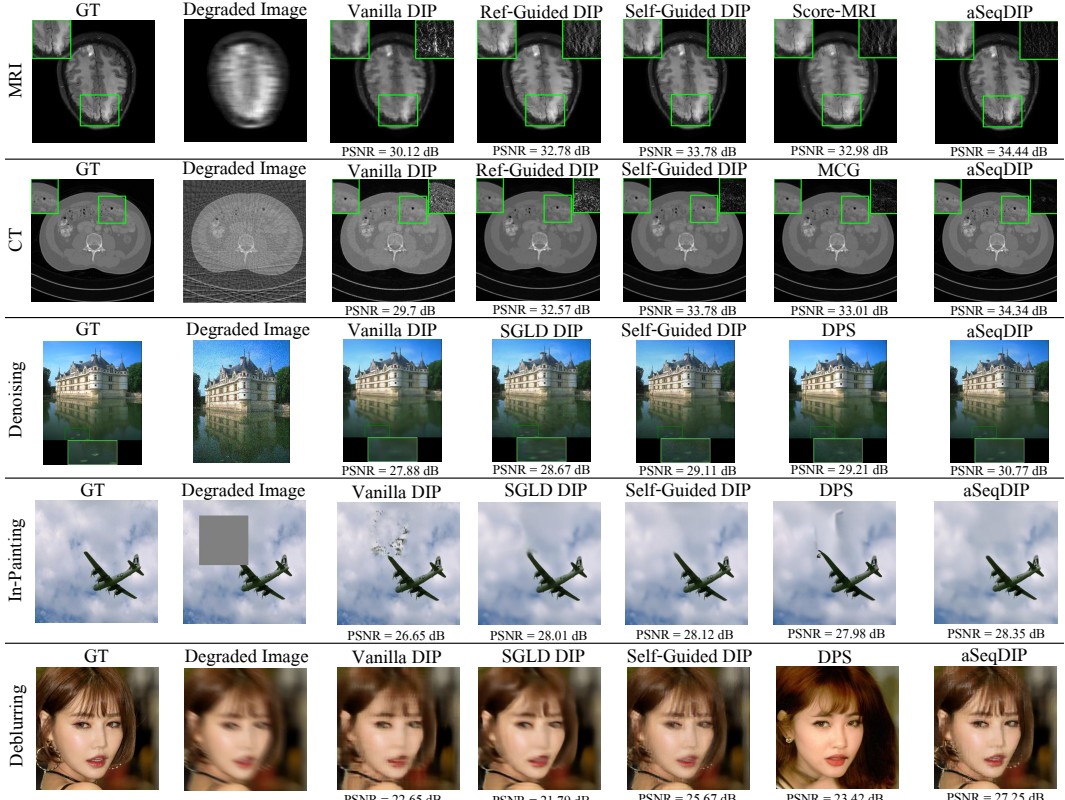

Figure 5: Reconstructed/recovered images using our proposed approach, aSeqDIP, and the baselines for the considered tasks. The ground truth (GT) and degraded images are shown in the first and second columns, respectively, followed by three or four baselines per task. The last column presents our method. PSNR results are given at the bottom of each reconstructed image. For MRI (8x undersampling) and CT (18 views), the top right box shows the absolute difference between the center region box of the reconstructed image and the same region in the GT image. Denoising and in-painting used $\sigma_d = 25$ and HIAR = 0.25. For the task of Deblurring, aSeqDIP contains artifacts when compared to DPS. However, DPS generates a perceptually different image when compared to the GT. For all other tasks, aSeqDIP reconstructions contain sharper and clearer image features than other methods.

# 5    Conclusions & Future Work

In this paper, we introduced Autoencoding Sequential Deep Image Prior (aSeqDIP), a new unsupervised image recovery algorithm. Notably, aSeqDIP operates without the need of pre-trained models, relying solely on a sequential update of network parameters. These parameters are optimized using an input-adaptive data consistency objective combined with autoencoding regularization, effectively mitigating noise overfitting. Our experimental results across various tasks highlight the competitive performance of the proposed algorithm, matching (or outperforming) diffusion-based methods in terms of reconstruction quality and required run time, all without the need for pre-trained models.

For future directions, we aim to explore the applicability of aSeqDIP to other image recovery problems, thereby expanding its versatility and potential impact across diverse domains. Additionally, we are interested in investigating the integration of a network input update mechanism to dynamically adjust the autoencoding regularization parameter and the number of gradient updates per iteration.

# Acknowledgments

This work was supported by the National Science Foundation (NSF) grants CCF-2212065, CCF-2212066, and BCS-2215155. The authors would like to thank Xiang Li (University of Michigan), Avrajit Ghosh (Michigan State University), Huijie Zhang (University of Michigan), and Siddhant Gautam (Michigan State University) for insightful discussions.

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

# Appendix

In the Appendix, we first shed more light on the impact of the DIP network input by studying the training dynamics using the neural tangent kernel for CNNs with residual connections. Next, we show how trained autoencoders on clean images can be used as reconstructors at testing time. Lastly, we provide additional experimental results and visualizations.

## A Case Study: Impact of the DIP Network Input through lens of Neural Tangent Kernel in Residual Networks

We show the impact of the DIP input through the lens of the Neural Tangent Kernel (NTK) [30, 44] for residual networks[6]. The NTK is a tool used to analyze the training dynamics of neural networks in the infinite width limit, where for CNNs the network width corresponds to the number of channels. In this limit, the change of any individual parameter during training becomes very small, which means that the change in the network's output during training can be accurately approximated by a first order Taylor expansion around its initialization. In the context of DIP, we consider training a neural network $f$ with parameters $\theta$ and a fixed input $\mathbf{z}$ using gradient descent. At each training iteration, the network parameters are updated according to:

$$\theta^{(t+1)} = \theta^{(t)} - \beta \nabla_\theta \mathcal{L}(f_{\theta^{(t)}}(\mathbf{z})), \tag{7}$$

where $\mathcal{L}$ is the loss function, and $\beta$ is the learning rate. We also consider the resulting change in the network's output due to this parameter update using the first order Taylor expansion:

$$f_{\theta^{(t+1)}}(\mathbf{z}) \approx f_{\theta^{(t)}}(\mathbf{z}) + \nabla_\theta f_\theta(\mathbf{z})\big|_{\theta=\theta^{(t)}}(\theta^{(t+1)} - \theta^{(t)}). \tag{8}$$

Substituting (7) into (8) and applying the chain rule to write

$$\nabla_\theta \mathcal{L}(f_{\theta^{(t)}}(\mathbf{z})) = (\nabla_\theta f_\theta(\mathbf{z})\big|_{\theta=\theta^{(t)}})^T (\nabla_{f_{\theta^{(t)}}(\mathbf{z})} \mathcal{L}(f_{\theta^{(t)}}(\mathbf{z}))) \tag{9}$$

yields the equation:

$$f_{\theta^{(t+1)}}(\mathbf{z}) \approx f_{\theta^{(t)}}(\mathbf{z}) - \beta \underbrace{(\nabla_\theta f_\theta(\mathbf{z})\big|_{\theta=\theta^{(t)}})(\nabla_\theta f_\theta(\mathbf{z})\big|_{\theta=\theta^{(t)}})^T}_{\mathbf{\Theta}^{(t)}} (\nabla_{f_{\theta^{(t)}}(\mathbf{z})} \mathcal{L}(f_{\theta^{(t)}}(\mathbf{z}))) . \tag{10}$$

In the infinite width limit, NTK theory states that the matrix $\mathbf{\Theta}^{(t)} := (\nabla_\theta f_\theta(\mathbf{z})\big|_{\theta=\theta^{(t)}})(\nabla_\theta f_\theta(\mathbf{z})\big|_{\theta=\theta^{(t)}})^T$ stays fixed throughout training, so that $\mathbf{\Theta}^{(t)} = \mathbf{\Theta}^{(0)}$ for all $t$. This matrix is called the neural tangent kernel, and we denote it as $\mathbf{\Theta}$. Moreover, because the parameters $\theta^{(0)}$ are initialized randomly, in the infinite width limit, the NTK $\mathbf{\Theta}$ becomes deterministic (as a function of $\mathbf{z}$) due to the law of large numbers [30], and does not depend on the specific instantiation of $\theta^{(0)}$. In DIP, the loss function is the least squares loss given in (1). For simplicity, we consider the denoising case, where the forward operator $\mathbf{A} = \mathbf{I}$. Then, substituting the gradient of the loss into (10) shows explicitly how the output of deep image prior evolves during training:

$$f_{\theta^{(t+1)}}(\mathbf{z}) = f_{\theta^{(t)}}(\mathbf{z}) + \beta \mathbf{\Theta}(\mathbf{y} - f_{\theta^{(t)}}(\mathbf{z})). \tag{11}$$

Using this recursion relation, one can derive a closed form of the network output at iteration $t$ in terms of the initial output and NTK [20, 30]. The reconstruction at iteration $t$ is given by:

$$f_{\theta^{(t)}}(\mathbf{z}) = \mathbf{y} - (\mathbf{I} - \beta \mathbf{\Theta})^t(\mathbf{y} - f_{\theta^{(0)}}(\mathbf{z})). \tag{12}$$

It is evident from (12) that the initial reconstruction of the network, $f_{\theta^{(0)}}(\mathbf{z})$, has important effects on the training dynamics of DIP. Furthermore, networks used in DIP often feature skip connections from earlier layers to later ones, and it is natural to believe that these connections may cause the input $\mathbf{z}$ to have a large effect on $f_{\theta^{(0)}}(\mathbf{z})$. In the following theorem, we analyze the training dynamics of CNNs with a very similar architectural modification: a residual connection that adds the input directly to the network output.

---

[6]We note that skip and residual connections are not exactly the same as skip represents concatenation (typically from encoder to decoder end) and residual represents adding the input to the output. However, both operations correspond to sending initial input or features of a network to its latter portion or output.

**Theorem A.1** (Dynamics of DIP with Residual Connections). *Let $g$ be a convolutional neural network with parameters $\theta$. We consider the complementary residual network $f$ defined by $f_\theta(\mathbf{z}) = \mathbf{z} + g_\theta(\mathbf{z})$. Suppose that $f$ is trained using gradient descent with the loss $\mathcal{L}(f_\theta(\mathbf{z})) = \frac{1}{2}||f_\theta(\mathbf{z}) - \mathbf{y}||_2^2$. Then, in the infinite width limit (number of channels), in expectation over the initialization of the parameters $\theta$, we have that the output at training iteration $t$ is given by:*

$$f_{\theta^{(t)}}(\mathbf{z}) = \mathbf{y} - (\mathbf{I} - \beta\mathbf{\Theta})^t(\mathbf{y} - \mathbf{z}). \tag{13}$$

The proof of Theorem A.1 is provided in Appendix A.1, along with a precise statement of the assumptions on the network architecture and parameter initialization. Additionally, in Appendix A.2 we provide a simple experiment to validate that the training dynamics given in (13) hold for real networks.

**Remark A.2.** *Theorem A.1 can be used to understand how the choice of network input affects the performance of DIP for image denoising. To gain intuition, we consider two special cases. First, we consider using the noisy image $\mathbf{y}$ as the input $\mathbf{z}$. In this case, equation (13) simplifies to $f_{\theta^{(t)}}(\mathbf{y}) = \mathbf{y}$ for all iterations $t$. In this case absolutely no denoising occurs. On the other hand, we consider the oracle case where the clean image $\mathbf{x}$ is used as $\mathbf{z}$. This gives us $f_{\theta^{(t)}}(\mathbf{x}) = \mathbf{y} - (\mathbf{I} - \beta\mathbf{\Theta})^t(\mathbf{y} - \mathbf{x})$. We see that at initialization ($t = 0$), we already expect perfect denoising, since $f_{\theta^{(0)}}(\mathbf{x}) = \mathbf{y} - (\mathbf{I} - \beta\mathbf{\Theta})^0(\mathbf{y} - \mathbf{x}) = \mathbf{x}$. These two cases support intuition that using a network input closer to the true image could result in better performance in fewer training iterations.*

## A.1 Proof of Theorem A.1

*Setting of Theorem A.1.* We first precisely state the conditions of Theorem A.1, in particular the network architectures considered and the corresponding parameter initializations. The present setting is very similar to the setting considered in [30], but we provide the details here for completeness. We consider an $L$ layer CNN with $c_{\text{in}}$ input channels and $c_{\text{out}}$ output channels, with $c$ hidden channels in all intermediate layers. We assume that $c_{\text{in}}, c_{\text{out}} << c$. We assume all convolutions have a filter size of $r$. For simplicity, the network input and output are vectorized, so convolutions of any dimension are treated identically. For example, for a 2D CNN with $5 \times 5$ kernels, $r = 25$. Written explicitly, a network $g$ with this architecture takes the form

$$g_\theta(\mathbf{z}) = C_L(\varphi(C_{L-1}(\varphi(\cdots \varphi(C_1(\mathbf{z})))))),$$

where the operators $C_i$ represent convolutions with an additive bias, and $\varphi$ is a pointwise activation function such as ReLU. In this section, we also consider the residual network architecture defined by $f_\theta(\mathbf{z}) = \mathbf{z} + g_\theta(\mathbf{z})$.

We assume that the parameters are initialized using the He initialization [45]. With this initialization, the first layer convolutional filter weights are drawn from $\mathcal{N}(0, \frac{\sigma_w^2}{c_{\text{in}}r})$, and the filter weights for all other layers are drawn from $\mathcal{N}(0, \frac{\sigma_w^2}{cr})$, where the variance $\sigma_w^2$ depends on the non-linearity used in the network. For ReLU networks, $\sigma_w^2 = 2$ [45]. All biases are initialized to 0.

*Proof.* In the setting described above, the NTK emerges in the limit $c \to \infty$. A body of existing theory [30, 44, 46] establishes that in this limit the NTK is a *deterministic* matrix as a function of the network input $\mathbf{z}$. This theory does not consider residual connections, but applies immediately to both $g$ and $f$. For $g$, the NTK is given by

$$\mathbf{\Theta} := (\nabla_\theta g_\theta(\mathbf{z})\big|_{\theta=\theta^{(0)}})(\nabla_\theta g_\theta(\mathbf{z})\big|_{\theta=\theta^{(0)}})^T. \tag{14}$$

However, we can see that $\nabla_\theta g_\theta(\mathbf{z})\big|_{\theta=\theta^{(0)}} = \nabla_\theta f_\theta(\mathbf{z})\big|_{\theta=\theta^{(0)}}$.

Therefore, the linearization given in equation (10) holds for $f$ using the same kernel $\mathbf{\Theta}$, and equation (12) describes the training dynamics of $f$.

Using equation (12), we can write:

$$f_{\theta^{(t)}}(\mathbf{z}) = \mathbf{y} - (\mathbf{I} - \beta\mathbf{\Theta})^t(\mathbf{y} - f_{\theta^{(0)}}(\mathbf{z})) \tag{15}$$

$$= \mathbf{y} - (\mathbf{I} - \beta\mathbf{\Theta})^t(\mathbf{y} - \mathbf{z} - g_{\theta^{(0)}}(\mathbf{z})) \tag{16}$$

To prove Theorem A.1, we consider the output $f_{\theta^{(t)}}(\mathbf{z})$ in expectation over the initialization $\theta^{(0)}$. Since all parameters $\theta^{(0)}$ are drawn from mean 0 gaussian distributions, we find that $\mathbb{E}_{\theta^{(0)}}[g_{\theta^{(0)}}(\mathbf{z})] = 0$ for any input $\mathbf{z}$. Since $\mathbf{\Theta}$ is deterministic in the limit $c \to \infty$, in expectation over $\theta^{(0)}$ equation (16) reduces to $f_{\theta^{(t)}}(\mathbf{z}) = \mathbf{y} - (\mathbf{I} - \beta\mathbf{\Theta})^t(\mathbf{y} - \mathbf{z})$, which proves Theorem A.1. $\square$

## A.2 Example to support the results of Theorem A.1

We now provide a simple example using real networks to support the validity of Theorem A.1. This experiment substantiates both of the special cases considered in Remark A.2. Additionally, it shows that using the ground truth as the network input greatly inhibits overfitting. Indeed, in this experiment, we find that a residual network trained with the ground truth as input takes approximately 25 times more training iterations to completely learn the noisy signal than a residual network trained using a random noise input.

We use DIP for denoising a 1D sinusoidal signal. We denote the clean signal $\mathbf{x}$. The noisy signal is $\mathbf{y} = \mathbf{x} + \mathbf{n}$, where $\mathbf{n} \sim \mathcal{N}(0, \mathbf{I})$. The network used is a five layer ReLU CNN with a residual connection. The full architecture can be written as $f(\mathbf{z}) = \mathbf{z} + C_5(\text{ReLU}(C_4(\cdots \text{ReLU}(C_2(\text{ReLU}(C_1(\mathbf{z}))))))))$, where each $C_i$ represents a convolution (with bias). The signal has a size of 100, and the convolutions each have a filter size of 3, with 64 hidden channels. In all cases, the network is trained using gradient descent with a learning rate of $5 \times 10^{-4}$. The same seed was used to initialize the network in all cases.

We trained this network using three different inputs: the true signal $\mathbf{x}$, the noisy signal $\mathbf{y}$, and noise $\mathbf{z} \sim \mathcal{N}(0, \mathbf{I})$. The results of training the network using these three inputs are shown in Figure 6. We find that the results for training this real, reasonably sized network show the behavior predicted by Theorem A.1, which was obtained in the infinite width limit. Indeed, equation (13) predicts that when $\mathbf{x}$ is used as the input, the initial error will be small, with eventual overfitting to the noisy signal. This is observed in Figure 6, where the error is lowest at initialization, and it steadily increases throughout training. With this input, the network is highly resistant to overfitting, requiring approximately $10^5$ training iterations to completely fit the noisy signal. We also see that when $\mathbf{y}$ is used as the input, the error curve obtained is essentially flat and converges quickly to the error of the noisy signal. This agrees with the expectation that the network output will be $\mathbf{y}$ for all iterations $t$ when $\mathbf{y}$ is used as the input. Finally, when random noise $\mathbf{z}$ is used as the input, the typical DIP behavior emerges.

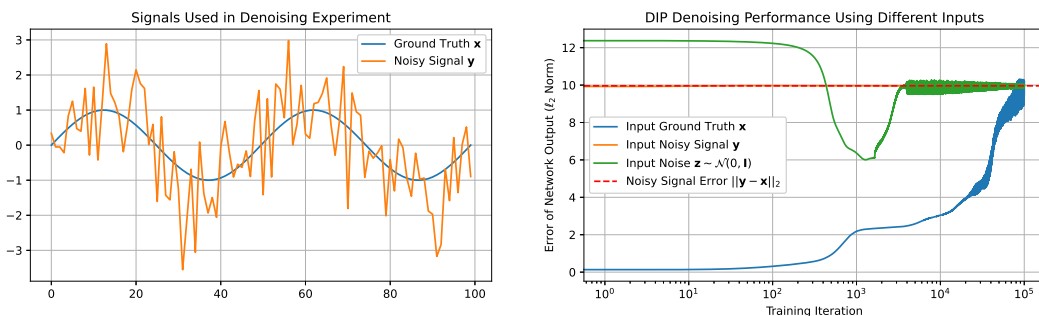

Figure 6: Ground truth signal and measurements (*left*), and the results of the denoising experiment in Appendix A.2 (*right*) to support the claims in Theorem A.1 and Remark A.2.

# B   Trained Autoencoders as Reconstructors

Here, we investigate how the autoencoder term in aSeqDIP is improving the reconstruction quality while mitigating the impact of noise overfitting. In particular, we try to answer the question: *Can an autoencoder trained on clean images operate as a reconstructor at testing time?*

To address this question, we perform the following steps: (*i*) train an autoencoder on fully sampled measurements or clean data images and (*ii*) utilize the trained autoencoder with unseen subsampled or corrupted measurements, optimizing over the input using the DIP objective with the autoencoder term. This enables the autoencoder to function as an image reconstructor. Specifically, given a training dataset, $\mathcal{D}$, comprising unperturbed images or fully sampled MRI/CT data, denoted by $\mathbf{x}$, we train an autoencoder U-Net $g : \mathbb{R}^n \to \mathbb{R}^n$ with parameters $\psi$. The training process seeks to obtain $\hat{\psi}$ as

$$\hat{\psi} = \operatorname*{argmin}_{\psi} \frac{1}{|\mathcal{D}|} \sum_{\mathbf{x} \in \mathcal{D}} \|g_\psi(\mathbf{x}) - \mathbf{x}\|_2^2 . \tag{17}$$

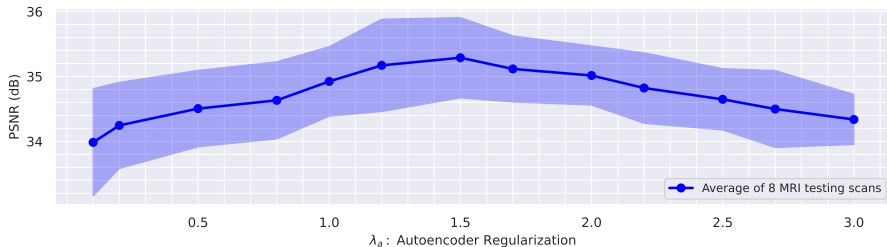

Figure 7: Average PSNR (y-axis) of 8 MRI images (with 4x undersampling) obtained by optimizing the input of a trained autoencoder (using (17)) w.r.t. different values of the regularization parameter $\lambda_a$ in (18) (x-axis).

Subsequently, given unseen measurements $\mathbf{y}$ and the learned autoencoder's parameters $\hat{\psi}$, we test the reconstruction of

$$\mathbf{z} \leftarrow \operatorname*{argmin}_{\mathbf{z}} \|\mathbf{A}g_{\hat{\psi}}(\mathbf{z}) - \mathbf{y}\|_2^2 + \lambda_a \|g_{\hat{\psi}}(\mathbf{z}) - \mathbf{z}\|_2^2 \,, \tag{18}$$

where $\lambda_a \in \mathbb{R}_+$ is a regularization parameter. We perform this experiment by training $\psi$ using 3000 fully sampled scans from the fastMRI dataset [47]. We then evaluate the reconstruction quality of the trained encoder using 8 scans from the fastMRI testing set. The average PSNR results for different values of $\lambda_a$ are depicted in Figure 7. As observed, a trained autoencoder effectively serves as a reconstructor as evidenced by the achieved PSNR. Thus, we deduce that the autoencoder term in aSeqDIP not only mitigates noise overfitting but also enhances the reconstruction quality as an important prior.

## C   Additional Experiments

### C.1   Robustness to Noise Overfitting for the Denoising Task

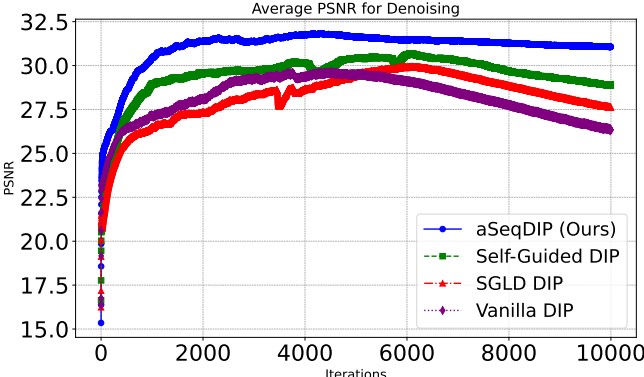

Figure 8: Average PSNR results of 20 images w.r.t. to the optimization iteration using aSeqDIP and DIP-based baselines.

In this subsection, we illustrate aSeqDIP's robustness to the noise overfitting issue for the denoising task. The average PSNR for 20 images from the CBSD68 dataset for denoising using aSeqDIP and other DIP-based methods are given in Figure 8. We observe two key points. First, in addition to higher PSNR, aSeqDIP shows higher robustness against noise overfitting compared to other DIP-based methods, consistent with MRI and CT results in Figure 4. Second, unlike MRI and CT, the onset of noise overfitting occurs earlier, but the subsequent decay is very small.

### C.2   Comparison with VarNet: An End-to-End MRI Supervised Method

Here, we compare aSeqDIP with the End-to-End (E2E) MRI supervised model [48] that uses the variational network (VarNet). Results are given in Table 3. As observed, we slightly under-perform when compared to VarNet (trained on 8000 data points) for the task of MRI reconstruction all without

requiring any labeled training data. It is important to note that, at inference, E2E models only require a few unrolling steps, whereas aSeqDIP is an optimization method that requires to train the network parameters for each new set of measurements.

| Task | Method | Data Independency | PSNR (↑) |
|---|---|---|---|
| MRI | E2E VarNet (trained on 8000 data points from fastMRI) [48] | × | **34.89** |
| | E2E VarNet (trained on 3000 data points from fastMRI) [48] | × | 33.78 |
| | aSeqDIP (Ours) | ✓ | 34.08 |

Table 3: Average PSNR results (over 20 MRI scans at 4x undersampling from the testing set of fastMRI) reported by our method against E2E VarNet [48] (pre-trained on fastMRI) for the task of MRI reconstruction.

## C.3 Comparison with DM-based Methods on the FFHQ Dataset

| Task | Method | Data Independency | PSNR (↑) |
|---|---|---|---|
| Denoising | DPS (trained on FFHQ) [16] | × | 31.45 |
| | DDNM (trained on FFHQ) [49] | × | 31.65 |
| | aSeqDIP (Ours) | ✓ | **31.77** |
| Random In-Painting | DPS (trained on FFHQ) [16] | × | 24.54 |
| | DDNM (trained on FFHQ) [49] | × | 25.54 |
| | aSeqDIP (Ours) | ✓ | **25.76** |
| Deblurring | DPS (trained on FFHQ) [16] | × | 23.67 |
| | DDNM (trained on FFHQ) [49] | × | 23.88 |
| | aSeqDIP (Ours) | ✓ | **24.02** |
| Box In-Painting | DPS (trained on FFHQ) [16] | × | 22.67 |
| | DDNM (trained on FFHQ) [49] | × | **22.89** |
| | aSeqDIP (Ours) | ✓ | 22.3 |

Table 4: Average PSNR results reported by our method against DPS as well as a more recent leading method DDNM [49] for four image restoration tasks: Denoising, Random In-Painting, non-linear Deblurring, and Box In-Painting.

Here, we present average PSNR results (averaged over 20 images) for the tasks of denoising , random inpainting (97% missing pixels), box-in-painting (with HIAR of 0.25), and non-linear deblurring of our method versus Denoising Diffusion Null-Space Model (DDNM) [49] and DPS [16] on the FFHQ testing dataset. For DPS and DDNM, we used a pre-trained model that was trained on the training set of FFHQ. As observed, our training-data-free method achieves competitive or slightly improved results when compared to data-intensive methods on all tasks other than box-inpainting (for which we under-perform by less than 1 dB), all without requiring a pre-trained model.

## C.4 Ablation Study on the Regularization Parameter in aSeqDIP

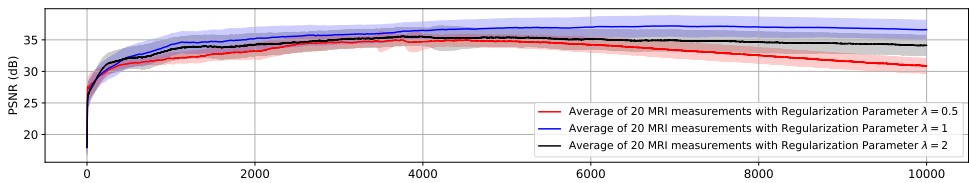

Figure 9: Average PSNR results of 20 MRI (with 4x undersampling) scans in aSeqDIP for the cases where $\lambda \in \{0.5, 1, 2\}$ and $i \in [10000]$.

In this section, we conduct an ablation study on the choice of the autoencoding regularization parameter, $\lambda$, in (3). Specifically, we conduct an experiment using 20 MRI scans to examine the impact of $\lambda$ on the reconstruction quality and noise overfitting in aSeqDIP. We note that while the main results in Section 4 use $NK = 4000$, in these experiments, we run our algorithm for an extended number of iterations to investigate the onset of noise overfitting. We set $K = 5000$ and $N = 2$ for this purpose.

In this experiment, we run aSeqDIP with values of $\lambda \in \{0.5, 1, 2\}$. Average PSNR results are given in Figure 9. It is evident that, on average, using $\lambda = 1$ yields the most favorable results in terms of

PSNR values, which is our selected choice. Furthermore, we observe that for $\lambda = 0.5$ (red), the start of the PSNR decay (the onset of noise overfitting) precedes that of $\lambda = 1$ and $\lambda = 2$ (blue and black).

## C.5 Ablation Study on $N$ and $K$ in aSeqDIP

In this section, we conduct an ablation study to investigate the impact of the number of gradient updates ($N$) per one set of parameter ($K$) in aSeqDIP. Specifically, we report the PSNR results across the tasks of MRI, CT, denoising, and in-painting for the case of $NK = 4000$, considering combinations of $(N, K)$ as $(1, 4000)$, $(2, 2000)$, and $(4, 1000)$. The results, presented in Figure 10, reveal that across all tasks considered, the combination of $N = 2$ and $K = 2000$ consistently yields the most favorable results in terms of PSNR.

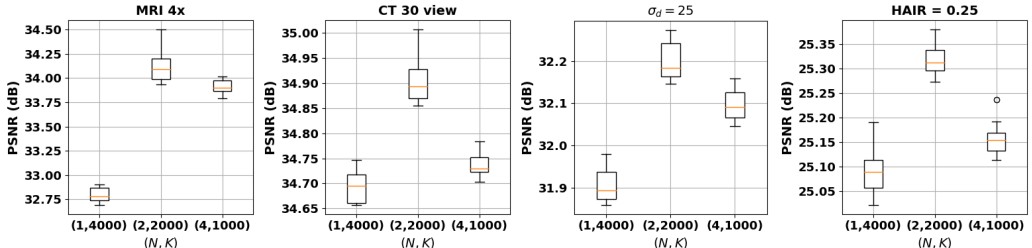

Figure 10: Ablation study on the choice of the number of gradient updates $N$ per U-Net, and the number of U-Nets $K$ in terms of PSNR using the four considered tasks for the case of $NK = 4000$.

## C.6 Additional Implementation Details

For denoising, each noisy RGB image is $512 \times 512$ and is generated by adding an additive Gaussian white noise with two noise levels as descired in Table 1. For in-painting, we consider a central region mask, and we evaluate two hole-to-image area ratios (HIAR) with image size $512 \times 512$.

For Vanilla DIP, Self-Guided DIP, Reference-guided DIP, TV-DIP, Rethinking DIP, and SGLD DIP, we use 4000 iterations. For TV-DIP, we set the regularization parameter to 1. For ES-DIP, we use the default configuration provided by the authors for the three considered tasks.

The reference image in Reference-guided DIP is chosen as (using a distance metric such as Euclidean distance or other metric) that which is most similar to an estimated test reconstruction from undersampled data or sparse-view data.

For DM-based approaches, we use the codes attached to the authors' papers. Specifically, Score-MRI [14], MCG [15], and DPS [16]. For our experiments with natural images (denoising, in-painting, and non-linear deblurring) in Table 2, we used the CBSD68 dataset. As such, for DPS (the DM-based method), we utilized a pre-trained model that was trained on a very large and diverse dataset which is ImageNet $128 \times 128$, $256 \times 256$, and $512 \times 512$. This pre-trained model is much more generalizable when compared to the other option which was trained on FFHQ (a dataset of faces). According to [17], the ImageNet pre-trained model has high generalizability. For the FFHQ comparison results in Appendix C.3, we used an FFHQ-pre-trained DM for DPS and DDNM.

For MRI, the pre-trained model used in Score-MRI was originally trained on natural images then fine-tuned using the training set of fastMRI. Similar approach was used for the CT pre-trained model used in MCG.

## C.7 Additional Visualizations

Figures 11 and 12 present additional MRI visualisations, whereas Figures 13 and 14 present CT visualizations. Samples from the natural image restoration tasks are given in Figure 15, Figure 16, and Figure 17 for box-inpainting, denoising, and non-linear deblurring, respectively.

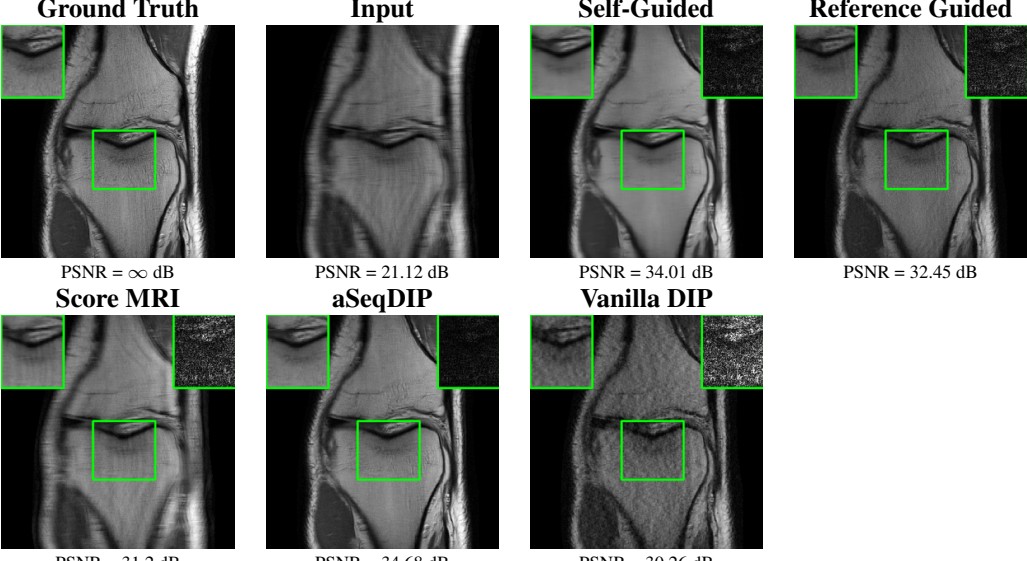

Figure 11: Visualization of ground-truth and reconstructed images using different methods of a knee image from the fastMRI dataset with 4x k-space undersampling. A region of interest is shown with a green box and its error (magnitude) is shown in the panel on the top right. aSeqDIP provides the sharpest and clearest reconstruction of image features.

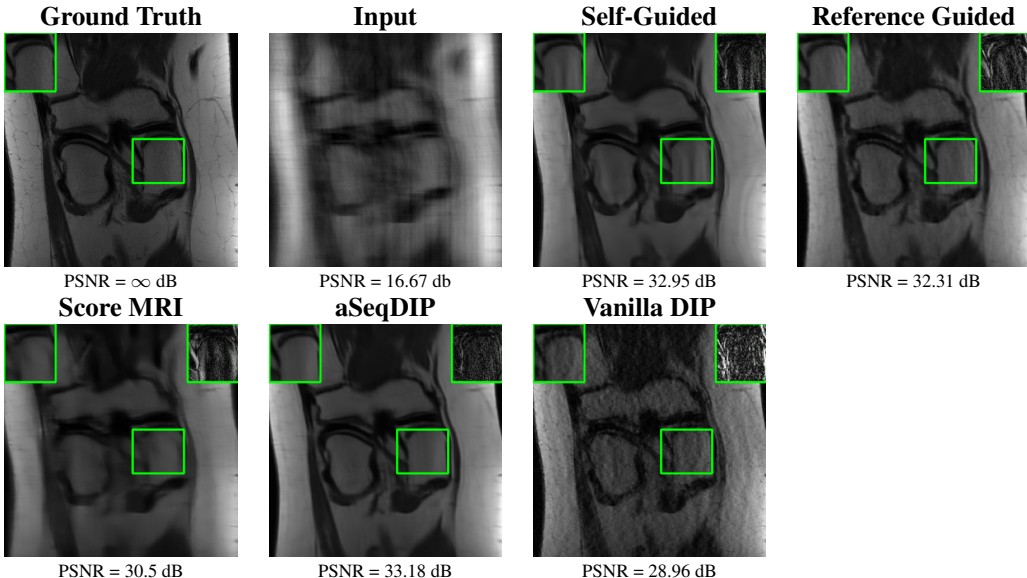

Figure 12: Visualization of ground-truth and reconstructed images using different methods of a knee image from the fastMRI dataset with 8x k-space undersampling. A region of interest is shown with a green box and its error (magnitude) is shown in the panel on the top right. aSeqDIP provides the clearest reconstruction of image features amongst the methods.

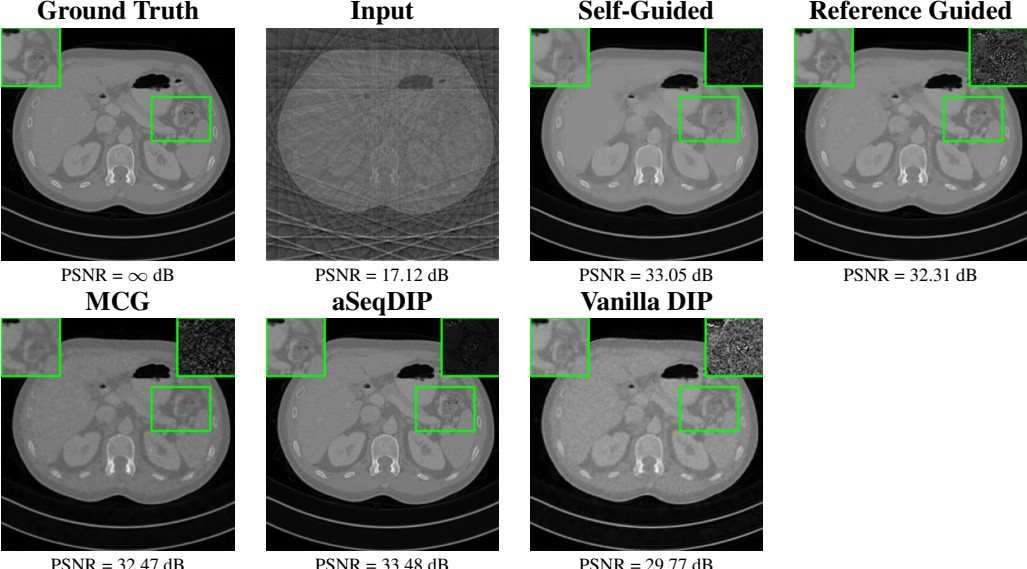

Figure 13: Visualization of ground-truth and reconstructed images using different methods of a CT scan from the AAPM dataset with 18 views. A region of interest is shown with a green box and its error (magnitude) is shown in the panel on the top right. aSeqDIP provides the sharpest and clearest reconstruction of image features.

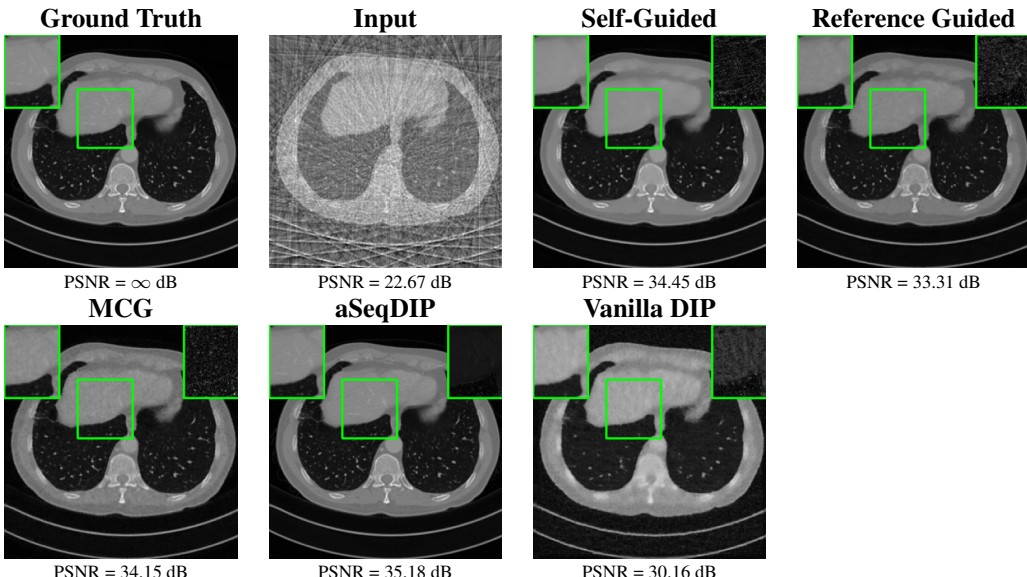

Figure 14: Visualization of ground-truth and reconstructed images using different methods of a CT scan from the AAPM dataset with with 30 views. A region of interest is shown with a green box and its error (magnitude) is shown in the panel on the top right. aSeqDIP provides better reconstruction of small and low-contrast image features.

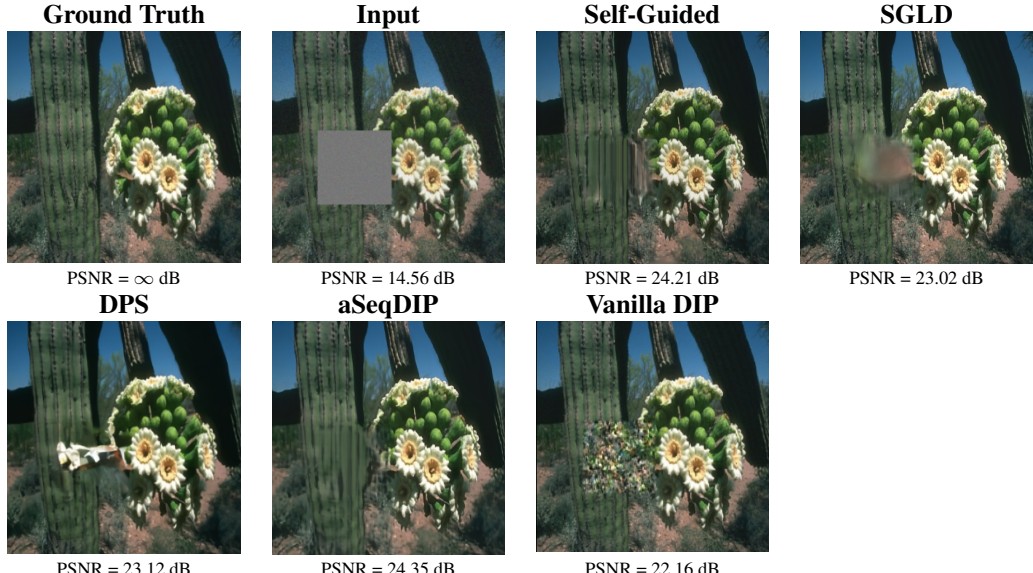

Figure 15: In-painting example with 0.1 HIAR where image restorations of different methods are given using an example from the CBSD68 dataset. The diffusion-based DPS produces spurious (although sharp) content in the hole region while aSeqDIP much better preserves features in the original ground truth.

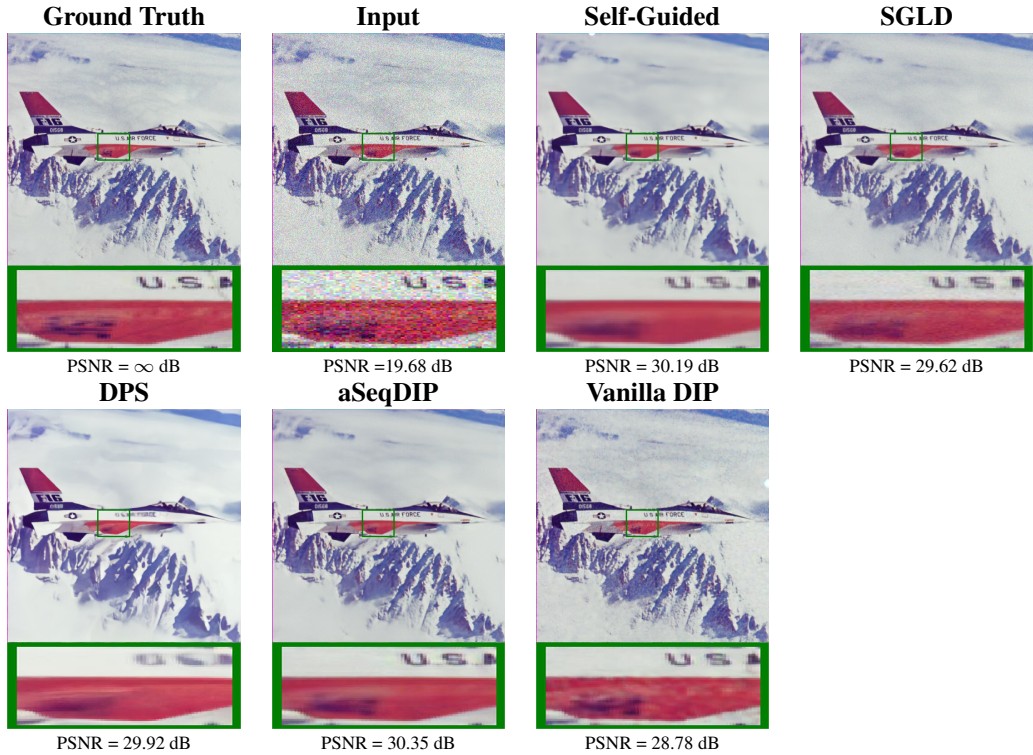

Figure 16: Denoising example with $\sigma_{\mathrm{d}} = 25$ where image restorations of different methods are given using an example from the CBSD68 dataset. aSeqDIP provides the sharpest and clear reconstruction of image features.

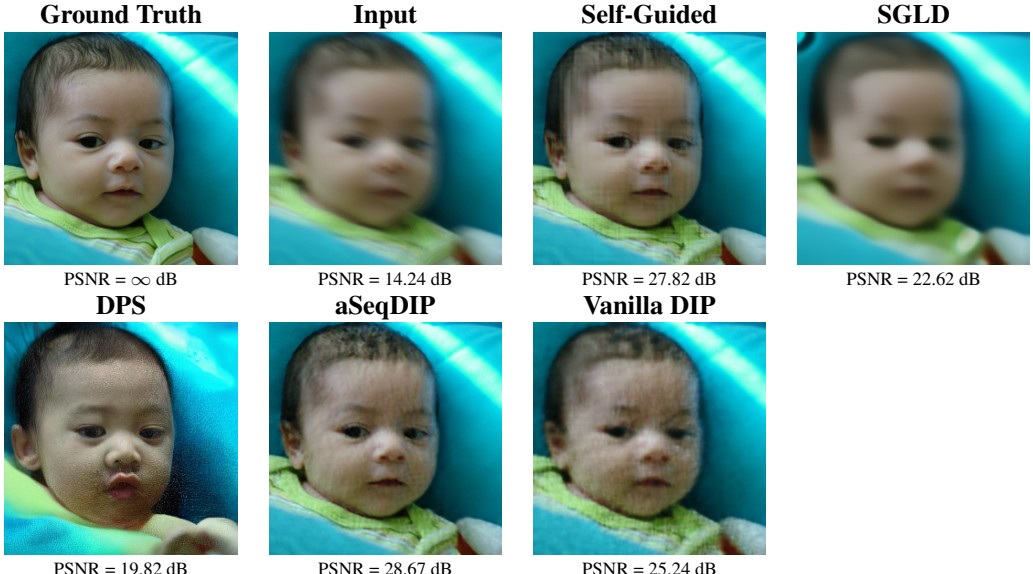

Figure 17: Non-linear deblurring samples where image restorations of different methods are given using an example from the FFHQ dataset.

