# OpenReview forum: "Image Reconstruction Via Autoencoding Sequential Deep Image Prior"
_NeurIPS.cc/2024/Conference — NeurIPS 2024 poster_

### Official Review · Reviewer_Q1ff · 2024-07-08

**Soundness:** 3
**Presentation:** 2
**Contribution:** 3
**Rating:** 6
**Confidence:** 4

**Summary:**

The paper proposes Autoencoding Sequential Deep Image Prior (aSeqDIP) an extension to the family of untrained (no training data required) DIP methods.
As in previous work a combination of data consistency loss and autoencoding regularization (a loss between network input and output) is used. However, instead of further regularizing the optimization by adding noise to the network input in every iteration as proposed in previous work, this paper varies the network input by setting it to the network output after every two iterations.
Experimental evidence is provided that demonstrates superior robustness to overfitting and overall reconstruction accuracy compared to other DIP approaches and trained diffusion models.

**Strengths:**

Significance: Untrained neural networks are an important technique used in many fields, where training data is scarce. Mitigating the problem of overfitting is an important contribution.

Originality: Despite the relatively small difference compared to previous work (regularize network input by setting it equal the output instead of adding noise to it as in the self-guided DIP) the importance and effectiveness of this idea seems to be justified by the experimental results in terms of reconstruction accuracy and robustness to overfitting.

Clarity: Overall the paper is written in a clear way and related work in terms of DIP approaches and diffusion models are discussed.

Quality: The experiments towards evaluating the reconstruction performance and overfitting robustness of the proposed DIP approach are well designed and ablation studies for the most important hyperparameters are provided.

Overall an interesting paper. However, some questions remain and some important points remain unclear (see weeknesses). If those can be addressed adequately, I would consider raising the score.

**Weaknesses:**

**1. No results are provided for the most interesting case: Overfitting robustness for the task of denoising.**
To my understanding Prop. 3.1 says that in the limit the proposed approach converges to a network that has learned the identity and to a network output (which is identical to the network input) that perfectly fulfills data consistency under the forward operator and the given measurement.
For the tasks of MRI and inpainting (in the absence of noise) solutions that fulfill Prop. 3.1 include the ground truth image and the empirical evidence provided in the paper shows that indeed the solution found by the proposed method is closer to the ground truth image than that of other DIP based approaches (why it is closer remains unclear up to intuitions/speculations, but answering this question is non-trivial and does not have to be the scope of the paper).

Also, the robustness to overfitting seems reasonable for the aforementioned tasks. My intuition would be that repeatedly setting the network input to the output combined with the autoencoding regularization loss lets the network converge to identity relatively quickly. If at the same time data consistency is reached, there is no incentive for the network to change its weights anymore and a stable solution is reached.

**However, things change in the presence of noise or in general for the task of denoising.**
To my understanding, for denoising Prop. 3.1 is fulfilled, when the network perfectly reconstructs the noisy measurement $\mathbf{y}$, which is the very definition of overfitting. So, here it would be interesting to see curves as presented in Figure 4 for MRI and CT also for denoising to demonstrate to what extend overfitting is still prevented.

**2. In general, I am not sure about the usefulness of Prop. 3.1.**
As discussed above for the tasks of MRI and inpainting reconstructions that fulfill Prop 3.1 comprise good and bad solutions however without making a statement why a good solution should be preferably reached over a bad solution, whereas for the task of denoising it implies convergence to a bad solution.

**3. The learning based "SOTA" baselines perform very badly, which is not discussed in the paper and no information regarding the used training sets is provided.**
Diffusion model-based methods are introduced as the state-of-the-art in the paper (line 113) and indeed the works referenced there report impressive results in terms of quantitative and qualitative reconstruction performance.
Yet, in this work the reconstruction scores and especially the reconstructed images (Figure 5, 12, 13) contain severe artifacts. I understand that training and inference of those diffusion models is sensitive to hyperparameters, but then potential reasons for this bad performance should at least be discussed in the paper. Also, no information is provided regarding the training data used for those baseline models.

If there is no comparison to state-of-the-art end-to-end approaches (which definitely should give better results than DIP based reconstruction if trained properly on enough data) like the end-to-end VarNet (https://arxiv.org/abs/2004.06688) then at least the diffusion model based baseline should show reasonable results or if not it should be discussed why.

**4. Code is only provided for the task of MRI reconstruction and the implemented method seems to differ from the description in the paper.** Checking the code provided via a link (https://anonymous.4open.science/r/Aseq_DIP-E728/README) in the paper I can only find a notebook that performs MRI reconstruction *Aseq_DIP.ipynb*.
In the code the step where the network output is set to be the network input for the following steps is implemented as follows:
```
    pred_ksp = mps_and_gt_to_ksp(mps1.to(device), net_output)

    new_pred_ksp = (1 - mask_from_file).to(device) * pred_ksp.detach() / scale_factor + mask_from_file * ksp1

    new_ref = ksp_and_mps_to_gt(new_pred_ksp, mps1)
    random_smoothing_temp = torch.zeros_like(new_ref).to(device)

    for jj in range(1):
        random_smoothing = new_ref + 1e-5 * torch.rand((640, 372), dtype=torch.complex64).to(device)
        random_smoothing_temp += random_smoothing

    random_smoothing_final = random_smoothing_temp
    ref[:, 0, :, :] = random_smoothing_final.real
    ref[:, 1, :, :] = random_smoothing_final.imag
```
If I understand it correctly the new input (ref) is first set to fulfill perfect data consistency and then processed with random smoothing, two things not mentioned anywhere in the paper. Also, the new input (ref) is updated before the computation of the loss, which means that in fact the autoencoding regularizer is not really computed between the network input and the network applied to this same input.
It is unclear how essential these two steps (hard data consistency and smoothing) are to the proposed approach.

Further,
```
    loss.backward()
    for i in range(2):
        optimizer.step()
```
implies that the network weights are updated twice without updating the loss functions, which is not what is outlined in Algorithm 1 of the paper.

If that is really how the mehtod is implemented that corresponds to the results in the paper the ablation study regarding the values for the number of iterations $N$ per network in Appendix C.4 seems meaningless as the loss is not updated as $N$ increases.

**Questions:**

1. In the Figures (like Fig. 5) showing qualitative reconstruction results it would be helpful to show the measurement, i.e. noise image for denoising, mask fr in-painting and zero-filled reconstruction for MRI.

2. I personally find the formulation in line 10 "sequential optimization of multiple network architectures" a bit missleading as the networks are not changed or re-initialized. Only the network input is adapted.

3. Regarding Section 3.1 and Figure 2 it makes sense to me why the blue curve is decaying, but do you know why the red curve is decaying? Does the red curve decaying imply that a constant input (like all zeros) to the DIP would work best?

**Limitations:**

Limitations are not discussed in the paper. This could be for example the limitted interpretability of the comparison to trained baseline methods as current results in the paper significantly fall behind the results reported in the baseline papers or the long reconstruction times compared to end-to-end deep learning based methods.

---

> ### Author Rebuttal · Authors · 2024-08-07
>
> **C1 Overfitting robustness for the task of denoising**: Since our method is an unsupervised approach optimizing a single image, our observations suggest that repeatedly setting the network input to the output, along with autoencoding regularization, enables convergence to autoencoding-specific images instead of the identity.
>
> The attached PDF in the global response presents the average PSNR for 20 images from the CBSD68 dataset for denoising. We observe two key points. First, aSeqDIP shows higher robustness against noise overfitting compared to other DIP-based methods, consistent with MRI and CT findings. Second, unlike MRI and CT, the onset of noise overfitting occurs earlier, but the subsequent decay is very small. Please refer to our response to the following comment for the proposition.
>
> **C2 Usefulness of Prop. 3.1**: The main point of this proposition is to highlight that the aSeqDIP algorithm minimizes a different optimization problem compared to Vanilla DIP. Proposition 3.1 is not intended to indicate solution quality or analyze RMSE during optimization but to state that, under strong convergence assumptions, our algorithm minimizes the optimization problem in (5). Solving (5) explicitly is challenging due to the equality constraint and the network's non-linearity. In other words, the proposition illustrates the types of solutions our algorithm converges to. Our empirical results show that our algorithm converges quickly and remains stable.
>
> **C3 Performance of DM-based methods and their training data**: Regarding the difference in PSNR values between our reported results and those in Score-MRI and MCG, we'd like to clarify that for our MRI experiments, we used Cartesian sampling, a more common scheme, while Score-MRI used Uniform 1D, Gaussian 1D, Gaussian 2D, and VD Poisson disk sampling. For CT, our aSeqDIP results (and other baselines except MCG) were obtained using the full 512x512 pixels in the AAPM dataset, whereas MCG downsized the images to 256x256 pixels for faster training and sampling (see the caption of Table 3 in the MCG paper). Therefore, we compared the ground truth after resizing the MCG results back to 512x512. We will include this discussion in the revised paper. In the following table, we compare aSeqDIP (a dataless method) versus MCG (a data-centric method) at their downsized pixel space, as reported in Table 3 of their paper. As observed, we achieve very competitive results.
>
> | Task | MCG 256X256 | aSeqDIP 256X256 |
> |----------|----------|----------|
> | Sparse View CT with 18 views | 33.75 | 33.86 |
> | Sparse View CT with 30 views | 36.09 | 35.89 |
>
> We use the testing images from the fastMRI and AAPM datasets. The pre-trained models for Score-MRI and MCG were trained on the training sets of these datasets.
>
> For our experiments with natural images (denoising, in-painting, and non-uniform deblurring), we used the CBSD68 dataset. For DPS (a DM-based method), we utilized a pre-trained model from ImageNet (128x128, 256x256, and 512x512), known for its high generalizability according to "The Emergence of Reproducibility and Consistency in Diffusion Models." This model is more generalizable than the alternative trained on FFHQ (faces dataset). In response to Reviewer MAra Comment 4, we experimented with the FFHQ testing set, using pre-trained models of DPS and DDNM trained on the FFHQ training set.
>
> For the DM-based baselines, we used the default hyper-parameters provided by DPS (natural images), Score-MRI, and MCG (sparse view CT).
>
> Note that we are not the first to observe unwanted artifacts in DPS (see Figure 3 in "Solving Inverse Problems with Latent Diffusion Models via Hard Data Consistency"). We believe these artifacts arise because DM-based approaches, which learn $p(x)$, require modifications to sample from $p(x | y)$. These modifications, often approximations, may not be accurate or suitable for all tasks and images.
>
> A major point of our comparison is to show that our method achieves competitive or superior results compared to DM-based methods, without needing pre-trained models or training data.
>
> End-to-end (E2E) supervised models outperform many DIP-based methods like VarNet because the best DIP optimization steps can vary significantly. This motivates our aSeqDIP approach. The table below shows that our method achieves higher average PSNR than VarNet for 15 MRI scans (4x). While E2E models are faster at inference with a single forward pass, our method's advantage is that it is fully data-independent.
>
> | Task | VarNet | aSeqDIP (Ours) |
> |----------|----------|----------|
> | MRI | 33.78 | 34.08 |
>
> **C4 Code**: Thank you for bringing this to our attention. We mistakenly uploaded an older version of our code. We have fixed the link and uploaded .py files for all the tasks.
>
>
> **Q1 Including measurements**: See our global response.
>
> **Q2 Comment on using "sequential optimization of multiple network architectures"**: In our algorithm, each network is initialized by the optimized parameters of the previous network, and then optimized using (3) such that the input of the network is the output of the previously optimized network (see line 157). No re-initialization or architecture change are needed. In the revised manuscript, we will re-word it for further clarification.
>
> **Q3 Curves of Figure 2**: Larger variance in the standard Gaussian distribution corresponds to larger additive perturbations even for the case of $\mathbf{x}^*=\mathbf{0}$ (the red curve). We conjecture that this still leads to larger distances from the GT and hence worst performance. In regard to the case where the input is the all-zero vector, we ran Vanilla DIP with $\mathbf{z}=\mathbf{0}$, and the reconstruction quality is very low. We conjecture that this is due to (i) the impact of the first convolutional layer, which is only its bias, and (ii) the output of the first layer is concatenated to later layers through the skip connection.
>
> **Limitation Comment**: See our global response.

---

> > ### Comment · Reviewer_Q1ff · 2024-08-09
> > **Reponse to author rebuttal - remaning concerns with baseline results**
> >
> > I thank the authors for the clarifications and additional experiments and for answerning my qustions.
> >
> > My remaining concerns are with the baseline results. If Score-MRI is so unstabel that when trained on the entire fastMRI brain training set is still outperformed by a data-free method on the fastMRI brain validation set, then maybe it is not a good DM baseline. Also the type of undersampling mask should not make up for the difference in PSNR as Uniform 1D and Gaussian 1D are also Cartesian masks.
> >
> > Further, I find it extremely strange that the VarNet is outperformed by a data-free method as the VarNet provides stable SOTA results (see fastMRI challenge) if trained and tested on the same type of data.
> > So what was the training and testing setup for this experiment with the VarNet?

---

> ### Author Response · Authors · 2024-08-10
> **Thank you for your response. Addressing the remaining concerns with baselines results**
>
> We would like to thank the reviewer for their prompt response to our rebuttal. We hope that the following will address the reviewer's remaining concerns with baselines.
>
> We would like to emphasize that the main point of comparing our method with data-intensive approaches (VarNet, Score-MRI, DPS, DDNM, and MCG) is to demonstrate that we achieve competitive results, all without the need for training data or pre-trained models by appropriately setting up the optimization and regularization of deep image prior. In what follows, we address your concerns in three parts.
>
> **If Score-MRI is so unstable that when trained on the entire fastMRI brain training set is still outperformed by a data-free method on the fastMRI brain validation set, then maybe it is not a good DM baseline.**
>
> Score-MRI DM was originally trained on natural images and then fine-tuned on the entire training set of fastMRI using the single-coil setting. The reason for this pre-training + fine-tuning is that DMs require a significant amount of training data to enter the generalization regime (see Figure 2b in "The Emergence of Reproducibility and Consistency in Diffusion Models"), which may not be available for tasks such as MRI and CT. During testing, they used the multicoil real setting DM on both the real and imaginary parts. In their paper, they mention: "Our model requires magnitude images only for training, and yet is able to reconstruct complex-valued data, and even extends to parallel imaging."
>
> To the best of our knowledge, Score-MRI and CCDF (which we came across after submitting the paper) are the two best DM-based MRI baselines. In CCDF, they demonstrated slightly improved PSNR scores while requiring fewer sampling steps, making it faster. Refer to Table 5 in the CCDF paper, "Come-Closer-Diffuse-Faster: Accelerating Conditional Diffusion Models for Inverse Problems through Stochastic Contraction." In that paper, Score-MRI is referred to as "Score-POCS." For the 4x Gaussian 1D sampling, CCDF reports 32.51 dB, whereas Score-MRI achieved 31.45 dB.
>
> **The type of undersampling mask should not make up for the difference in PSNR as Uniform 1D and Gaussian 1D are also Cartesian masks.**
>
> Thank you for your comment. Note that in the Score-MRI paper (multicoil setting in Table 2), the PSNR values vary by nearly 4 dB depending on the mask, ranging from 29.17 dB (Gaussian 2D) to 34.25 dB (Gaussian 1D).
>
> We acknowledge the reviewer's comment that the masks they used are indeed Cartesian. Thank you for the correction. We needed to review the details of their code implementation to confirm this.
>
> The first column of Figure 4 (and its caption) in their paper describes the exact masks they used. Our sampling mask is the 1D Poisson Disk, which the Score-MRI authors did not use. The Poisson Disk mask in their results is 2D Poisson Disk.
>
> To fully address the reviewer's concern, we ran aSeqDIP with the 1D Uniform setting, the sampling mask used in the first row of Table 2 in Score-MRI. The results are averaged over 20 MRI knee scans (with 4x undersampling). As observed, we achieve competitive PSNR results with Score-MRI in this setting as well.
>
> | Task | Score-MRI (reported from their paper) | Score-MRI (us running their code)| aSeqDIP (Ours) |
> |----------|----------|----------|----------|
> | MRI (1D Uniform Sampling) | 33.25 | 33.45 | 34.05 |
>
>
> **Further, I find it extremely strange that the VarNet is outperformed by a data-free method as the VarNet provides stable SOTA results (see fastMRI challenge) if trained and tested on the same type of data. So what was the training and testing setup for this experiment with the VarNet?**
>
> The reviewer is correct; VarNet indeed achieves very competitive results. Since VarNet does not provide a pre-trained model, we initially trained their architecture from scratch using 3,000 fastMRI multicoil datapoints and tested it with the fastMRI testing dataset. Please note that due to time constraints and the additional experiments conducted during the rebuttal, in our initial response, we used 3,000 knee images for training instead of the available 8,000 datapoints.
>
> To fully address the reviewer's concern, we trained VarNet with the full training set and their PSNR results do indeed improve when compared to the results of VarNet trained with only 3K points. See the following table. Our method is still quite competitive and could prove beneficial in limited training data regimes.
>
> | Task |	VarNet (trained with 8k points)	| VarNet (trained with 3k points) | aSeqDIP (Ours) |
> |----------|----------|----------|----------|
> |MRI	| 34.89 | 33.78	 | 34.08 |
>
>
> Note that Score-MRI paper also reported VarNet results (7th column of Table 2), and the results slightly varied when different masks were used.
>
> *We are happy to address any more concerns*.
>
> Thanks,
>
> Authors

---

> > ### Comment · Reviewer_Q1ff · 2024-08-11
> > **Some more details regarding the VarNet experiment?**
> >
> > Thank you for providing the additional information regarding the DM baseline experiments. I guess Score-MRI looses more performance through this shift from training on magnitude images only to multi-coil evaluation than I expceted. Still impressive that your aSeqDIP can perform on the same or better level.
> >
> >
> > Regarding the VarNet experiment, I have a last question.
> > If I understand it correctly, you consider the problem of multi-coil knee slice reconstruction.
> > Do you focus on a certain subset of slices or which data do you mean when you say 3000 out of the 8000 available fastMRI multi-coil datapoints?
> > The fastMRI knee dataset contains alsmot 35k slices, see https://arxiv.org/pdf/1811.08839 Table 4.

---

> > > ### Author Response · Authors · 2024-08-11
> > > **Thank you for your response. Response to the VarNet experiment**
> > >
> > > We would like to thank you again for your response. We are glad that you found the results of aSeqDIP impressive. We hope that our responses can further convince you to raise the score.
> > >
> > > We double checked and the reviewer is correct that the full training/validation set is larger. We would like to clarify the settings we used for the rebuttal. We used a subset of data and removed peripheral slices in each volume (around 10 per volume) during training. We followed similar setup as recent works "Blind Primed Supervised (BLIPS) Learning for MR Image Reconstruction, TMI 2021" (Fig. 3) that showed supervised model results with varying training sizes from ~1K to ~8K. We believe its promising that a data-free approach can compete with supervised networks trained with many knee slices.

---

> > > > ### Comment · Reviewer_Q1ff · 2024-08-12
> > > > **Thanks for the clarification. Final summary of revision.**
> > > >
> > > > Thanks for the clarification. I trust that in the revised version all the additional experimental settings and baseline configurations are explained in detail at least somewhere in the Appendix.
> > > >
> > > > Overall I am very happy with the rebuttal and the additional experiments and clarifications provided by the authors.
> > > > In particular, the experiment on robustness towards overfitting in the denoising case demonstrates the benfits of the proposed approach, and the experiments with the end-to-end VarNet help to put the results into the right perspective.
> > > >
> > > > I raised my score accordinly.

---

> ### Author Response · Authors · 2024-08-12
> **Thank you for your response and raising your score**
>
> We would like to thank the reviewer for their response and raising their score. We are glad that the reviewer is satisfied with our rebuttal and the additional experiments.
>
> Following the reviewer's recommendation, we will add the additional experiments and more details about the baselines configurations in the revised manuscript.

---

### Official Review · Reviewer_hjHc · 2024-07-12

**Soundness:** 3
**Presentation:** 2
**Contribution:** 3
**Rating:** 7
**Confidence:** 4

**Summary:**

This manuscript describes a variant of the seminar Deep Image Prior (DIP) work that incorporates aspects and mindset of the iterative prediction workflow in diffusion models.
In particular, the proposed procedure (method?) aSeqDIP is training a network to predict a single and fixed but distorted (e.g. noisy) image when given a pixel-wise random input image (just as DIP has initially proposed). The key novelty is to only train for a few training steps before switching to feeding the current prediction as an updated input (instead of the initial noisy input image). The similarity to predictions with diffusion models is apparent.
The approach is to the best of my knowledge novel and the idea, in my point of view, the biggest contribution of the paper.

**Strengths:**

* The idea is fantastic and thought provoking.
* Introduction of the regularization term (autoencoding term) during iterative training and showing that it is useful and that tuning it is important. (Figure 7 from the supplement could make it into the main paper.)
* Application to 4 different tasks (MRI and CT (important real-world use-cases) and denoising and in-painting (to make the CV community happy… ;)
* Partially informative appendix.

**Weaknesses:**

* It surprises me that the manuscript does not argue more about WHY this approach leads to better results even when compared to data-dependent baselines. How strong are these baselines? Do better pre-trained methods exist? If so, why not show results with them as well?
   * An answer to the above “WHY?” question would go right to the heart of why diffusion works well and would therefore be very interesting.
   * Since the presented method/procedure is not dependent on any amount of available training data (but the single compromised target image) I wonder how this manuscript can avoid talking about inductive biases of the used network architecture (a UNet, hence, a CNN).
* The manuscript is overall rather compact in content, the text certainly not too dense but maybe even a bit too repetitive and wordy, and the figures not very legible (adequately sized). I also much regret not to see the compromised target image used by aSeqDIP, but only GT and final predictions.
* Figure 1: when seeing it first it really did not help me understand the paper any better. After reading the entire manuscript and coming back to Figure 1 I can confirm the figure makes sense, but one would hope that Figure 1 is more educational as it currently is.
* Figure captions are in general not terrible but also not in all cases making the figures self-contained. It is necessary to find some potentially distant place in the manuscript to fully grasp the visuals (e.g. Fig. 2).

**Questions:**

* I cannot judge the quality or sufficiency of the used baseline methods. If for any of the 4 tasks the current SOTA (or industry standard) method would also be given and compared against, I would find that very useful.
* When does the proposed approach work (when is it applicable) and what are known limitations of its applicability? (When does the inductive bias of the setup not suit the desired task?)

**Limitations:**

* I would expect that the inductive biases that comes with the used network and training procedure dictate for what problems the presented method/procedure can produce good results. If the authors have any thoughts, I think the manuscript would much benefit from a short discussion in that vein.

---

> ### Author Rebuttal · Authors · 2024-08-07
>
> **C1 WHY this approach achieves better results? Strength of the DM-based baselines. Inductive bias discussion**: We thank the reviewer for their constructive comment. The answers to these questions will definitely strengthen our paper. We divided our response to this comment into the following three parts.
>
> - Answer to why we achieve better results when compared to diffusion models: Our approach and DM-based methods are conceptually different. Provided a set of training images, DMs are trained to approximate the underlying distribution $p(x)$. Subsequently, when employed to solve imaging inverse problems (IPs), the problem becomes sampling from the conditional distribution $p(x\mid y)$ where $y$ denotes the measurements. To this end, DM-based IP solvers attempt to approximate this conditional distribution (which may not be accurate and/or suitable for some tasks) using different approaches for which the reverse sampling steps are modified to achieve this target. aSeqDIP is different in the sense that it is an optimization-based method that depends solely on an input-adaptive Unet architecture and the generative power of the network, and a loss function that is designed to mitigate irrelevant noise overfitting (which is the major obstacle with DIP-based methods). We argue that the reverse sampling modifications needed to sample from the conditional distribution in DM-based methods present a more challenging design choice when compared to our approach.
>
> - Answer to how strong the data-dependent baselines are: We believe that the DM-based methods that we used as baselines are very strong on the tasks they were considered for. For MRI and CT, we used the following criteria: Select a method that perform strongly for every task, and use the test dataset with the same training dataset distribution for which these DMs were trained on. For example, for MRI, we used Score-MRI as baseline. This method utilizes a pre-trained DM that was originally trained on natural images and then fine-tuned with the training set of the fastMRI dataset. In our MRI experiments, we used the testing set of fastMRI. Similar approach was used in CT where we used the MCG method (with an AAPM dataset-trained DM) and the AAPM testing dataset.
>
> For our experiments with natural images (denoising, in-painting, and non-uniform deblurring which is added in this rebuttal), we used the CBSD68 dataset. As such, for DPS (the DM-based method), we utilized a pre-trained model that was trained on a very large and diverse dataset which is ImageNet 128X128, 256X256, and 512X512. This pre-trained model is much more generalizable when compared to the other option which was trained on FFHQ (a dataset of faces). According to `The Emergence of Reproducibility and Consistency in Diffusion Models', the ImageNet pre-trained model has high generalizability. In our response to Reviewer MAra Comment 4, we experimented with the testing set of the FFHQ dataset where the pre-trained models of DPS and DDNM (the DM-based baselines) were trained on the training set of FFHQ.
>
> The main message of our comparison with data-centric methods is to demonstrate that our method can achieve competitive or superior results compared to DM-based methods, all without requiring pre-trained models and training data.
>
> - Answer to why we generally achieve better results: We think that our method, similar to all DIP-based methods, benefits from the implicit bias inherited in the Unet architecture. The structure of a randomly initialized CNN is used as a prior as it was shown in the original DIP paper. The architecture of a generator network alone is capable of capturing a significant amount of low-level image statistics even before any learning takes place. However, the number of optimization steps required in DIP represents a challenge. *Therefore, we believe that our method, exploiting autoencoding regularization and input-updating, keenly taps into the generative and denoising nature of CNNs for more explicit regularization to alleviate overfitting*.
>
> **C2 Text is not too dense but a bit too repetitive and wordy. Including compromised target images**: We do agree that some points are sort of repeated in the Introduction and other places in the paper. In the revised manuscript, we will improve the writing and readability of the figures. In the PDF attached in the global response of this rebuttal, we included the degraded images in our visualizations of Figure 5. We will do the ones in the Appendix in the revised manuscript.
>
> **C3 Figure 1 location and explanation**: We acknowledge the reviewer's comment. In the revised manuscript, we will either elaborate more in the caption of this figure (or add more context to the diagram itself) or re-locate it until after we present our method.
>
> **C4 Figures captions**: Thank you for your comment. We will improve the captions in the revised manuscript.
>
> **Q1 Sufficiency of the used baseline methods**: To address your question about how strong the DM-based baselines are, please refer to our response to your first comment. See also the second part of our response to comment 3 of Reviewer Q1ff where we experimented with a leading end-2-end supervised reconstruction model. Regarding the DIP-based baselines, we would like to highlight that we considered the recent self-guided DIP work which has demonstrated highly competitive performance in terms of reconstruction quality and robustness to noise overfitting across multiple tasks.
>
> **Q2 known limitations of aSeqDIP**: Thank you for your question. Please see our global response. Additionally, we evaluate our approach using different additional tasks, baselines, and settings. We hope that these results will shed more light into the capabilities and limitations of our method. In particular, for run-time and practical convergence, see our response to comment 3 of Reviewer qnNG. For testing with a non-linear task, see our response to Reviewer MAra comments 2 and 4.

---

> ### Author Response · Authors · 2024-08-12
> **A friendly and gentle reminder**
>
> We would like to express our sincere gratitude to the reviewer once again for their insightful comments.
>
> As the open discussion period is drawing to a close, we would be deeply grateful if the reviewer could kindly respond to our rebuttal. This would provide us with the opportunity to address or clarify any remaining concerns thoroughly.

---

### Official Review · Reviewer_MAra · 2024-07-13

**Soundness:** 2
**Presentation:** 3
**Contribution:** 1
**Rating:** 3
**Confidence:** 4

**Summary:**

The authors in this paper propose an Autoencoding Sequential DIP (aSeqDIP) which aims to address the overfitting issue of DIP while without introducing extra parameters. The idea is very simple, the authors simply feed the output of the DIP into DIP model after each N updates. The authors validate the efficiency of their methods on several different image restoration tasks.

**Strengths:**

1) The presentation of the paper is good and it is very easy to understand.
2) The authors have conducted experiments on for different image restoration tasks.
3) The authors provide numerical comparisons as well as visual comparisons.

**Weaknesses:**

I have several concerns about this paper.

1) The novelty is significantly limited. It is almost the original DIP. The only difference is: the original DIP each iteration uses a random noise as its input; while here, the output of DIP is fed into the DIP for the next N iteration's updates.

2) The current experiments are all linear image restoration tasks. It would be interesting to see how this model works for non-linear image restoration tasks such as image delurring.

3) The authors should compare their method with more advanced DIPs such as Ref1.

4) How does this model compare with SOTA diffusion models Ref2?

5) The current results (Table2), the improvement is very small. And I am wondering what if we run the experiments many rounds and report the mean and std.


Ref1: Jo, Yeonsik, Se Young Chun, and Jonghyun Choi. "Rethinking deep image prior for denoising." Proceedings of the IEEE/CVF International Conference on Computer Vision. 2021.

Ref2: Wang, Yinhuai, Jiwen Yu, and Jian Zhang. "Zero-shot image restoration using denoising diffusion null-space model." arXiv preprint arXiv:2212.00490 (2022).

**Questions:**

I have several questions which have been listed in [Weaknesses].

**Limitations:**

I have listed the limitations in [Weaknesses].

---

> ### Author Rebuttal · Authors · 2024-08-07
>
> **C1 Novelty and Differences with the original DIP**: We appreciate the reviewer's comment. We would like to emphasize that our proposed method differs significantly from other DIP-based methods, including Vanilla DIP. These distinctions, which we believe set our work apart, are highlighted below.
>
> - Formulation and Algorithmic Perspective: In addition to the input-adaptive nature of Algorithm 1 (which is motivated by the discussion in Section 3.1), we propose the use of the auto-encoding term (Section 3.2.1) which is not used in Vanilla DIP. Furthermore, the original DIP does not require the input to be an image, whereas in our case, it is an image-to-image mapping.
> - Noise Overfitting Perspective: The proposed input-adaptive procedure along with the auto-encoding term result in the major key benefit of significantly delaying the noise overfitting decay which is the main challenge in all of the DIP-based approaches. See Figure 4, Figure 10, and the first figure in the PDF attached in the global response of this rebuttal.
> - Applicability Perspective: In our paper, we presented experimental results using multiple reconstruction and restoration tasks. Most other DIP-based methods consider at most one to three tasks. For example, ref-guided DIP only considered MRI, whereas TV-DIP considered denoising and in-painting. Furthermore, in addition to the four tasks in the paper, in this rebuttal, we included a non-linear inverse imaging task which is non-uniform deblurring (see our response to the following comment).
>
> Based on the highlighted remarks, we respectfully ask the reviewer to reconsider their opinion regarding novelty.
>
>
> **C2 It would be interesting to see how this model works for non-linear image restoration tasks such as image deblurring**: We thank the reviewer for their comment. Here, we include results of the non-linear non-uniform image deblurring task using the setting in the DPS code. In particular, we use the ''blur-kernel-space-exploring'' setting. In what follows, we report the achieved PSNR (averaged over 25 images form the CBSD68 dataset) of aSeqDIP when compared to DPS, Self-guided DIP, and SGLD-DIP. As observed, our method significantly outperforms all DIP-based methods while reporting improved results when compared to DPS.
>
> | Task | DPS | Self-Guided DIP | SGLD-DIP | aSeqDIP (Ours)
> |----------|----------|----------|----------|----------|
> | Non-uniform Deblurring | 23.4 | 20.3 | 19.8 | 23.89 |
>
> **C3 Comparison with "Rethinking DIP for denoising"**: Thank you for your comment. In what follows, we compare aSeqDIP with the suggested paper. We use the task of denoising and report the average PSNR over 25 images from the CBSD68 dataset. As observed, our method reports higher PSNR. We believe that our approach, exploiting autoencoding regularization, keenly taps into the generative and denoising nature of CNNs for more explicit regularization to alleviate overfitting.
>
> | Task | Rethinking DIP for Denoising | aSeqDIP (Ours)
> |----------|----------|----------|
> | Denoising | 30.98 | 31.51 |
>
> **C4 Comparison with SOTA DM-based method, DDNM**: Thank you for your question. In what follows, we present average PSNR results (averaged over 20 images) for the tasks of denoising (with $\sigma_d=25$), random in-painting (97\% missing pixels), box-in-painting (with HIAR of 25), and non-uniform deblurring of our method versus DDNM  (the suggested paper) and DPS on the FFHQ testing dataset. For DPS and DDNM, we used a pre-trained model that was trained on the training set of FFHQ. As observed, our training-data-free method achieves competitive or slightly improved results when compared to data-intensive methods on all tasks other than box-inpainting (for which we slightly under-perform), all without requiring a pre-trained model.
> | Method | Denoising | Random In-painting | Non-uniform Deblurring | Box In-painting
> |----------|----------|----------|----------|----------|
> DDNM (using FFHQ-trained DM) | 31.45 | 25.34 |23.88 | 22.89|
> DPS (using FFHQ-trained DM) | 31.65 | 25.54 | 23.67 | 22.67|
> aSeqDIP (Ours) | 31.77 | 25.76 | 24.02 | 22.3 |
>
> **C5 The current results (Table2), the improvement is very small**: Thank you for your comment. While the PSNR and SSIM improvements compared to other baselines are not very significant, we would like to highlight the following points. First, compared to DIP-based methods, our approach not only achieves higher reconstruction quality but also significantly improves robustness to noise overfitting. See the PSNR curves in Figures 4 and 10, as well as in the figure in the attached PDF in the global response. Second, when compared to DM-based methods, our approach not only achieves comparable or slightly improved PSNR and SSIM scores but also has the significant advantage of being independent of any training data and pre-trained models. We hope that emphasizing these points will highlight the additional advantages offered by aSeqDIP.
>
> Regarding running the experiments for many rounds, do you mean for our method or the other baselines? Or do you mean running our method with different initializations of $\phi_1$?

---

> > ### Comment · Reviewer_MAra · 2024-08-13
> >
> > Thanks to the authors for the detailed rebuttal. And it is glad to know that the authors added the experiments on nonlinear tasks. However, to 1) the image to image mapping in DIP is not new at all, this paper [ref1] also used the image as its input instead of random seed; 2) I do agree that DIP-based model suffers from the overfitting issues, however, there are several papers have addressed this issue, see ref2, ref3. 3) for the nonlinear tasks, the authors may want to compare with DIP-based deblurring models such as ref4.
> >
> >
> > Ref1: https://openaccess.thecvf.com/content_ICCVW_2019/papers/LCI/Mataev_DeepRED_Deep_Image_Prior_Powered_by_RED_ICCVW_2019_paper.pdf
> >
> > Ref2: https://arxiv.org/abs/2112.06074
> >
> > Ref3:https://arxiv.org/abs/2110.12271
> >
> > Ref4: https://openaccess.thecvf.com/content_CVPR_2020/html/Ren_Neural_Blind_Deconvolution_Using_Deep_Priors_CVPR_2020_paper.html

---

> > > ### Author Response · Authors · 2024-08-14
> > > **Thank you for your response**
> > >
> > > We're glad the reviewer is satisfied with our testing of aSeqDIP on a non-linear task. Below, we address the remaining concerns.
> > >
> > > ### 1) The image to image mapping in DIP is not new, [ref1: DeepRED] also used the image as its input instead of random seed.
> > >
> > > In our rebuttal, we outlined the key differences between our work and the original DIP paper. We agree with the reviewer that we are not the first to consider a DIP network input containing some structure of the ground truth, **as discussed in line 88 of our paper, where we cite two other works**. Below, we discuss how aSeqDIP differs from [Ref1].
> > >
> > > We agree that DeepRED initializes the algorithm with the noised image $x_0 = y$. However, in DeepRED, the DIP network input is still random noise that remains fixed. This is stated in Section 4 as ***"In all the reported tests the same network as in [Original DIP] is used with an i.i.d. uniform (∼[0, 0.1]) random input tensor of size 32×W×H, where W×H is the size of the output image to synthesize"***. This input is $z$ in their algorithm which remains unchanged as given in Eq.(7), (11), and (12).
> > >
> > > In our case, the DIP network input is the image we are iteratively estimating, whereas for DeepRED, its a random tensor unrelated to the reconstruction.
> > >
> > > Additionally, we see three major differences between our work and DeepRED:
> > >
> > > - **Variables**: Due to their adoption of the ADMM algorithm, DeepRED requires updating three variables: The network parameters, variable $x$, and the Lagrange multipliers vector. In aSeqDIP, we only update the parameters and the input (updated by a single pass of the network after every few parameters update).
> > >
> > > -  **External Denoiser**: Due to the use of RED (The Little Engine that Could: Regularization by Denoising), in addition to the DIP network, $\textrm{T}_\Theta$, DeepRED requires an external denoiser $f$. $f$ is used for updating the network input (Eq.(11) or (12)). Table 1 in DeepRED shows the external denoisers (NLM and BM3D) used in their experiments. **In aSeqDIP, no external denoisers is needed**.
> > >
> > > - **Applicability**: We considered a diverse array of tasks including two medical image reconstruction tasks (MRI and CT) and three natural image restoration tasks, whereas DeepRED considered three image restoration tasks.
> > >
> > > We will include this discussion in the revised manuscript.
> > >
> > >
> > >
> > > ### 2) I do agree that DIP-based model suffers from the overfitting issues, others paper addressed this issue. [ref2]: Early Stopping for Deep Image Prior, and [ref3]: Self-Validation: Early Stopping for Single-Instance Deep Generative Priors
> > >
> > > We would like to thank the reviewer for sharing these papers. As noise over-fitting is the major drawback of DIP-based methods, we agree that there are several works that address the noise overfitting issue including the ones we discuss (**and include as baselines**) such as (**TV-DIP, Ref-DIP, SGLD-DIP, & Self-Guided DIP**).
> > >
> > > **Due to time constraints, we are unable to compare with [ref2] and [ref3]. However, we will include discussions about [ref2,ref3] in the revised related work section as below**.
> > >
> > > In [ref2], the authors categorized DIP methods based on addressing noise over-fitting into: (i) Regularization, (ii) Noise modeling, and (iii) Early stopping (ES). The methods in [ref2] and [ref3] are both ES approaches, whereas aSeqDIP belongs to the "Regularization" category due to the use of the input-adaptive auto-encoding term. In [ref2], the authors use the running variance in Eq. (3) as the criteria for ES, whereas the authors of [ref3] propose combining self-validation and training to apply ES.
> > >
> > > Most importantly, we believe that aSeqDIP achieves high robustness to noise over-fitting. In Figure 10 ($\lambda = 1$), on average, noise over-fitting does not start until iteration 8000 with a subsequent very minimal decay for the task of MRI. The minimal decay in noise over-fitting is also observed in the PSNR curves for denoising in the attached PDF. In these experiments, we compared to SGLD-DIP, Self-guided DIP, and Vanilla DIP.
> > >
> > >
> > >
> > > ### 3) Comparison with DIP-based deblurring models such as [ref4]: Neural Blind Deconvolution Using Deep Priors
> > >
> > > In our rebuttal to the reviewer's comment (see **C2 It would be...**), we compared with two DIP-based methods (self-guided DIP and SGLD-DIP) for the task of non-uniform image deblurring.
> > >
> > > In an attempt to fully address the reviewer's comment, we ran the code of [ref4] using 20 FFHQ images, and report the results below.
> > >
> > >
> > > | Method | Forward Operator | Non-uniform Deblurring |
> > > |----------|----------|----------|
> > > DDNM (using FFHQ-trained DM)| Known |23.88 |
> > > DPS (using FFHQ-trained DM)|Known | 23.67 |
> > > SelfDeblur [ref4]|Unknown |  22.35 |
> > > aSeqDIP (Ours)| Known |  24.02 |
> > >
> > > While we achieve better results, we emphasize that [ref4] operates in a blind setting without access to the forward operator.
> > >
> > > **We hope that our responses have addressed the reviewer's concerns and kindly ask if they could reconsider their score.**

---

> ### Author Response · Authors · 2024-08-12
> **A friendly and gentle reminder**
>
> We would like to sincerely thank the reviewer once again for their insightful comments.
>
> As the open discussion period is drawing to a close, we would be deeply grateful if the reviewer could kindly respond to our rebuttal. This would provide us with the opportunity to address or clarify any remaining concerns thoroughly.
>
> We would also like to respectfully highlight that other reviewers have acknowledged the novelty and strengths of aSeqDIP, with remarks such as "The idea is fantastic and thought-provoking", "The proposed approach for enforcing DIP to find its fixed point is interesting and novel", "The introduction of the regularization term (autoencoding term) during iterative training and showing that it is useful and that tuning it is important", and "Robustness towards overfitting demonstrating the benefits of the proposed approach".

---

### Official Review · Reviewer_qnNG · 2024-07-16

**Soundness:** 3
**Presentation:** 3
**Contribution:** 2
**Rating:** 5
**Confidence:** 4

**Summary:**

This paper investigates how to prevent deep image prior (DIP) from overfitting to the noise or compressed measurements, which is a classic problem of DIP. To address the problem, the authors proposed Autoencoding Sequential DIP (aSeqDIP). The general idea of aSeqDIP is to cuts the overall training process into K sequential blocks, in each of which the input ($\mathbf{z_k}$) of the network is the previous output ($f(\mathbf{z_{k-1}})$), and the loss is defined as $\mathcal{L} = ||Af(\mathbf{z}_k) - y||_2^2 + || z_k - f(\mathbf{z}_k)||_2^2$, where the second term forces the output to be alike the input. As k increases, the network aims to find the network $f$ such that $z = f(z)$ and $||Af(\mathbf{z}_k) - y||_2^2$ is minimized. In a word, aSeqDIP forces the output to be a fixed point of the network that fits the measurements as much as possible.

**Strengths:**

1. The paper is well-written and easy to follow

2. The proposed approach for enforcing DIP to find its fixed point is interesting and novel

3. Superior performance has been demonstrated again current diffusion-model-based approaches for image reconstruction tasks.

**Weaknesses:**

1. Although the experiment is thorough in terms of the variety of image reconstruction tasks, I still think it misses an important baseline that uses explicit regularizer (e.g. TV) to stablize the learning process. It would strengthen the argument if the authors could include a comparison like that.

2. The current introduction is concise, but it does not convey the message that the key benefit of aSeqDIP is to address the problem of overfitting.

3. The authors do not discuss the limitations of the proposed method.

**Questions:**

1. Please consider including a baseline like DIP+TV in the experiment.

2. What is the runtime of aSeqDIP? A comparison between aSeqDIP and other baselines is preferred.

3. [Subjective] The proposition seems redundant: 1) it assumes very strong conditions (the convergence of the network), which, in my opinion distracts the reader from the key message; 2) it kinda shows how aSeqDIP prevent itself from overfitting, but it is not very clear without text. I suggest to write the proposition in the form of plain text with equations.

**Limitations:**

Please discuss the limitations such as runtime, practical convergence, etc.

---

> ### Author Rebuttal · Authors · 2024-08-07
>
> **C1+Q1: Comparison with TV-DIP**: We thank the reviewer for their comment. In what follows, we include a comparison with TV-DIP in terms of the average PSNR over 15 scans in MRI (with 4x and the fastMRI test dataset) and 20 images from the CBSD68 dataset for denoising and in-painting. As observed, our method outperforms TV-DIP.
>
> | Method | MRI | Denoising | In-painting |
> |----------|----------|----------|----------|
> | TV-DIP | 31.04 | 30.57 | 22.67 |
> | aSeqDIP (Ours) | 34.08 | 31.51 | 24.56 |
>
> **C2: The current introduction does not convey the message that the key benefit of aSeqDIP is to address the problem of overfitting**: We acknowledge the reviewer's comment. In the revised Introduction Section of the paper, we will emphasize that the main problem of DIP-based methods is noise overfitting as the selection of the number of optimization steps in DIP can differ not only from task to task but also from image to image within the same task. We will add a discussion that the main goal of introducing aSeqDIP is to mitigate the issue of noise over-fitting through the input-adaptive procedure and the use of the auto-encoding term.
>
> **C3 Discussing the limitations such as runtime and practical convergence**: Please see our global response.
>
> **Q2 Runtime of aSeqDIP**: We thank the reviewer for their comment. In the last column of Table 2, we report the average run-time (minutes) of all the methods. For the second best PSNR and SSIM (self-guided DIP), our method is 2X faster for MRI and CT reconstruction and requires 1 minute less than self-Guided DIP for denoising and in-painting. When compared to DM-based methods, aSeqDIP requires a slightly less run time while achieving an improvement in terms of the reconstruction quality (PSNR and SSIM). While DM-based methods only require function evaluations and our method is an optimization-based approach, the generally larger run-time reported for DM-based methods is due to the necessity of running a large number of reverse sampling steps.
>
> **Q3 [Subjective] The convergence assumption in the Proposition**: We agree with the reviewer that the convergence assumption in Proposition 3.1 is strong. The main point we'd like to convey in this proposition is that aSeqDIP is trying to solve the optimization problem in (5), which is different that Vanilla DIP. In Remark 3.2, we discuss this point by comparing aSeqDIP and Vanilla DIP. In the revised manuscript, we will present the point of the proposition as a remark for further clarification.

---

> > ### Comment · Reviewer_qnNG · 2024-08-12
> > **Thank you for the response**
> >
> > I thank the authors for their response, and they have addressed most of my concerns. One only question remaining is how the DIP+TV is finetuned? I suggest the authors to include a description on this in the camera-ready.

---

> ### Author Response · Authors · 2024-08-12
> **Thank you for your response. Addressing the Reviewer's Concern**
>
> First, we would like to thank the reviewer for their response.
>
> For TV-DIP, during the rebuttal, we used "https://arxiv.org/pdf/1810.12864" following the reviewer's suggestion. We coded the TV regularization term in Equation (3) and used Equation (5) to tune the parameters $\Theta$. According to Section 4.1 of the TV-DIP paper, we used 5000 optimization steps. For the architecture, we used the U-Net architecture (with skip connections) from the original DIP paper, which is the same as the one the authors describe in Figure 3.
>
> In regard to the hyper-parameters (including the regularization parameter), in Section 4.1, the authors of TV-DIP mentions: "All algorithmic hyperparameters were optimized in each experiment for the best signal-to-noise ratio (SNR) performance with respect to the ground truth test image".
>
> As the authors do not provide the regularization parameter, in the results we provided in the rebuttal, we used $\lambda = 1$, which is similar to aSeqDIP.
>
> To fully address the reviewer's concern and ensure that the choice of $\lambda$ we used (in our rebuttal) in TV-DIP is sufficient, we run TV-DIP with additional values of the regularization parameter for the task of denoising and report the average PSNR results for 20 images from the CBSD68 dataset. As observed, the TV-DIP results with $\lambda$ near 1 is better than other values we considered here. In all cases, aSeqDIP achieves better PSNR.
>
> | Task | TV-DIP ($\lambda = 0.1$) | TV-DIP ($\lambda = 0.8$) | TV-DIP ($\lambda = 1$)| TV-DIP ($\lambda = 1.2$) |  TV-DIP ($\lambda = 3$) |TV-DIP ($\lambda = 10$) | aSeqDIP (Ours) |
> |----------|----------|----------|----------|----------|----------|----------|----------|
> | Denoising | 30.02 | 30.54 | 30.57 | 30.61 | 30.43 |28.89 | 31.51 |
>
> Following the reviewer's suggestion, we will add this discussion and the TV-DIP results in the revised paper. We are happy to address any other concerns the reviewer might have.
>
> *We hope that, in light of our responses, the reviewer might consider raising their score.*
>
>
> Thanks,
>
> Authors

---

### Author Rebuttal · Authors · 2024-08-07

# Global Response

We thank the reviewers for their constructive comments. Many reviewers raised questions about the limitations and capabilities of the proposed approach, and requested experiments with different tasks, settings, and baselines. Below, we discuss these limitations and summarize the additional experiments and results obtained over the past week. These discussions will be included in the revised manuscript.

**Limitation Discussions**:

- Reviewer qnNG - Comment 3: We thank the reviewer for their comment. In terms of run time, we would like to point out that in Remark 3.4, we discuss the computational requirements of the proposed method, which is determined by (i) the $N\times K$ parameter updates, and (ii) the number of function evaluations necessary for updating the input of every network which is $K$. Furthermore, in Table 2, we include average run-time results of aSeqDIP as compared to the considered baselines. Furthermore, we have shown that our method is not that sensitive to the selection of $N$ and $K$ (along with $\lambda$) as, for the four considered tasks, we selected the same values. As our adoption of the autoencoding term delays the start of the PSNR decay, the empirical convergence plots for our method reaches nearly 5\% of its best possible PSNR at around iteration 2000 (see Figure 4).
- Reviewer hjHc - Question 2: A known limitation of our method is its slow run-time when compared to the inference time of End-to-End (E2E) supervised reconstruction models. However, it is important to note that our method, as it is a DIP-based approach, operates without any training data and pre-trained models.
- Reviewer Q1ff - Limitation Comment: Thank you for your comment. See our response to your third Comment that (i) justifies why the reported results in our paper and the DM-based baselines are slightly different, and (ii) includes additional aSeqDIP CT results on MCG's downsized pixel space. In regards to run-time, we agree with the reviewer that the inference of E2E methods is very fast when compared to our method and the DM-based methods. The reason is that E2E methods requires only one forward pass (or a few in the case of unrolling networks). However, it is important to note that our method, as it is a DIP-based approach, operates without any training data and pre-trained models.


**Summary of Additional Results**:
- MRI, Denoising, and In-painting PSNR results with aSeqDIP (Ours) vs. TV-DIP. (Reviewer qnNG).
- Non-uniform image Deblurring (a non-linear inverse imaging task) PSNR results of our method as compared with DPS, Self-guided DIP, and SGLD. (Reviewer MAra)
- Denoising PSNR results of our method as compared to Rethinking DIP. (Reviewer MAra)
- Denoising, non-uniform deblurring, and in-painting results of our method as compared with DDNM and DPS using the FFHQ testing dataset. (Reviewer MAra)
- CT reconstruction results of aSeqDIP as compared to MCG using the downsized 256X256 setting. (Reviewer Q1ff)
- MRI PSNR results of our method as compared with a supervised E2E method, VarNet. (Reviewer Q1ff)
- Average PSNR curves for the task of denoising with respect to iteration $i$ for aSeqDIP, Vanilla DIP, and Self-Guided DIP is given in the **attached PDF**. (Reviewer Q1ff)
- The degraded images in Figure 5 are given in the **attached PDF**. (Reviewers hjHc and Q1ff)

---

### Comment · Area_Chair_GG8K · 2024-08-09
**Rebuttal is online - please respond**

Dear Reviewers,

Authors carefully prepared their rebuttal - trying to address the concerns you have raised. Please check the rebuttals and join the discussion about the paper.

Regards,

---

### Decision · Program_Chairs · 2024-09-25

**Decision:**

Accept (poster)

**Comment:**

Authors present a novel DIP method that uses a sequence of DIPs akin
to diffusion models. The idea is backed up by theory and as far as I
can see, experimental results also support the idea.

Reviewers raised concerns regarding the experimental setup and as far
as I can see, authors addressed them as good as possible. Most
reviewers are happy with the rebuttal.

One reviewer raise concerns even after the rebuttal about comparisons
and experimental results. While these concerns have valid points, I
think the work is quite good.

Given the novelty in the article, supported by 3 reviewers, supportive
experimental results and the fact that it attained 1BA, 1WA and 1A, I
think the paper would be a great contribution to the conference.